# Potent neutralization by monoclonal human IgM against SARS-CoV-2 is impaired by class switch

Ilaria Callegari[1,2] (ID), Mika Schneider[1], Giuliano Berloffa[1] (ID), Tobias Mühlethaler[3], Sebastian Holdermann[1,2] (ID), Edoardo Galli[1,2] (ID), Tim Roloff[4,5], Renate Boss[6], Laura Infanti[7], Nina Khanna[8], Adrian Egli[4,5] (ID), Andreas Buser[7] (ID), Gert Zimmer[9,10,†] (ID), Tobias Derfuss[1,2,†] & Nicholas S R Sanderson[1,2,*,†] (ID)

## Abstract

To investigate the class-dependent properties of anti-viral IgM antibodies, we use membrane antigen capture activated cell sorting to isolate spike-protein-specific B cells from donors recently infected with SARS-CoV-2, allowing production of recombinant antibodies. We isolate 20, spike-protein-specific antibodies of classes IgM, IgG, and IgA, none of which shows any antigen-independent binding to human cells. Two antibodies of class IgM mediate virus neutralization at picomolar concentrations, but this potency is lost following artificial switch to IgG. Although, as expected, the IgG versions of the antibodies appear to have lower avidity than their IgM parents, this is not sufficient to explain the loss of potency.

**Keywords** antibodies; B cells; class switch; MACACS; membrane antigen
**Subject Categories** Immunology; Microbiology, Virology & Host Pathogen Interaction

## Introduction

Around 80% of the soluble antibody in blood is of the IgG class, with the remainder consisting mostly of IgM and IgA (Loh *et al*, 2013). Research attention and industrial development have both focused overwhelmingly on IgG, with the result that the class-specific properties of IgM are still very poorly characterized. One property that might make IgM particularly suitable as an early effector mechanism is the avidity advantage offered by its multi-meric (pentameric or hexameric) structure. It has been proposed that this can enable an IgM whose antigen-binding domain has a relatively modest affinity for antigen (as is typical in the early immune response, before affinity maturation has produced high affinity antibodies) to achieve avid binding to antigens with multiple identical epitopes that can simultaneously be bound by the antibody's 10 or 12 antigen-binding sites (Keyt *et al*, 2020). On the other hand, this mechanism presumably also enhances IgM binding to low affinity self antigens, which can explain their well-known predilection for self-reactivity (Nakamura *et al*, 1988). The penta-meric structure of IgM is also a strong stimulus for complement activation (Sharp *et al*, 2019), which can contribute to suppression of infection (Kurtovic & Beeson, 2021), but has the potential to exacerbate immunopathology (Polycarpou *et al*, 2020).

Understanding the class-dependent properties of antibodies requires monoclonal antibodies of various classes against a defined pathogen, but the isolation and characterization of pathogen-specific monoclonal IgM has lagged well behind the study of IgG, because of the technical difficulties in isolating antigen-specific B cells and in producing IgM antibodies recombinantly. In the SARS-CoV-2 pandemic, large numbers of people were infected with a novel pathogen, enabling the collection of blood samples containing pathogen-specific B cells of all classes, during the acute immune response. Having previously developed a suite of techniques suited for isolating IgM B cells specific for viral glycoproteins (Zimmermann *et al*, 2019), we had the opportunity to isolate and study naturally occurring antibodies in their original classes and assess the importance of class (i.e., Fc region) on antibody functional properties.

1 Department of Biomedicine, University of Basel and University Hospital Basel, Basel, Switzerland
2 MS Center, Neurologic Clinic and Policlinic, Research Center for Clinical Neuroimmunology and Neuroscience Basel (RC2NB), University Hospital Basel, University of Basel, Basel, Switzerland
3 Biophysics Facility, Biozentrum, University of Basel, Basel, Switzerland
4 Applied Microbiology Research, Department of Biomedicine, University of Basel, Basel, Switzerland
5 Clinical Bacteriology and Mycology, University Hospital Basel, Basel, Switzerland
6 Federal Food Safety and Veterinary Office, Bern, Switzerland
7 Regional Blood Transfusion Service, Swiss Red Cross, Basel, Switzerland
8 Infectious Diseases and Hospital Epidemiology, University Hospital Basel, Basel, Switzerland
9 Institute of Virology and Immunology, Bern & Mittelhäusern, Switzerland
10 Department of Infectious Diseases and Pathobiology, Vetsuisse Faculty, University of Bern, Bern, Switzerland
*Corresponding author. Tel: +41 61 265 2608; E-mail: nicholas.sanderson@unibas.ch
†These authors contributed equally to this work

## Results and Discussion

### Serum IgM is polyreactive

We collected serum and peripheral blood mononuclear cells (PBMC) from 34 donors after recovery from PCR-confirmed SARS-CoV-2 infection, 14 vaccinated donors, and 86 donors with no known history of SARS-CoV-2 infection or vaccination from April 2020 to June 2021. Demographic information about donors is presented in Appendix Table S1. We measured binding of serum IgM, IgG, and IgA to TE 671 rhabdomyosarcoma cells stably transfected with SARS-CoV-2 spike protein fused at the carboxyl terminus to mCherry (TE spike-mCherry) or untransfected control cells (TE 0). We used flow cytometry of antibody binding on antigen-expressing cells in addition to ELISA, to enable the assessment of nonspecific binding to normal cell surface proteins, and to improve detection of antibodies whose binding is dependent on membrane-inherent properties such as lateral mobility of antigens. SARS-CoV-2 spike-specific antibodies of the IgG and IgA classes increased in infected and vaccinated donors (Fig 1A).

IgM in serum from all unexposed or vaccinated donors bound in similar measure to spike-expressing and control cells, with a clear spike-specific signal in only 6 convalescent donors when tested by flow-cytometry. Since the presence of spike-specific IgM in donors at this time point after symptom onset is well established (Pickering et al, 2020; Zohar et al, 2020), the lack of a stronger IgM signal on the spike-expressing cells requires some other explanation, for example, nonspecific IgM binding to the non-antigenic cells, in line with the known polyreactivity of IgM in general (Nakamura et al, 1988). To test this explanation, we examined self-reactivity of IgM and IgG antibodies from donors before and after vaccination against SARS-CoV-2. As expected, sera from donors after vaccination showed increased IgG binding to spike-expressing cells, but no change in IgG reactivity to untransfected TE 0 cells. IgM, on the other hand, showed increased reactivity to both spike-expressing and untransfected cells (Fig 1B and C). An alternative explanation is that high affinity antibodies of other classes competitively displace the IgM. To test this hypothesis, we used an IgM capture ELISA, in which serum IgM is first immobilized on a plate, and then, spike-specific IgM detected with a labeled antigen (Fig 1D). Using this approach, all exposed donors clearly showed spike-binding IgM in serum. While the ELISA approach correctly identifies a higher fraction of exposed donors, the flow cytometric comparison between IgM binding to spike-expressing and non-expressing cells makes clear that some of the exposure-induced IgM binding activity is not virus-specific. Our data do not enable us to assess the quantitative tradeoffs between sensitivity and specificity, since this would require a more comprehensive cohort of control sera including donors exposed to other immunogenic pathogens. Infection with Epstein–Barr Virus, for example, has been shown to induce spurious IgM RBD ELISA reactivity (Pickering et al, 2020).

### SARS-CoV-2 spike-specific IgM B cells can be isolated by MACACS

The paucity of reports of high affinity, pathogen-specific IgM antibodies in the literature might reflect several factors. It is possible that such antibodies are not made and that the achievement of high affinity through somatic hypermutation and selection is always accompanied by class switch. Alternatively, the non-detection of these antibodies may have a technical cause, for example masking by abundant polyreactive IgM (Nakamura et al, 1988), or the relative difficulty of producing IgM recombinantly (Chromikova et al, 2015).

To address the potential masking of rarer virus-specific IgM B cells by the larger population of polyreactive IgM B cells, we adopted the strategy of membrane antigen capture activated cell sorting (MACACS) described by Zimmermann et al (2019). This technique identifies B cells whose affinity for a membrane protein extracellular domain antigen is high enough that they can extract the protein from the membrane and become activated in the process (Batista et al, 2001). Using capture of mCherry-tagged spike protein as a measure of antigen capture, and CD69 labeling as a marker of activation (Malinova et al, 2021), we assessed the phenotype of spike-capturing B cells in peripheral blood of 3 convalescent donors (Fig 2). Based on the expression of CD19, CD20, CD21, CD27, CD38, CD138, IgM, IgG, and IgA (Appendix Table S4), we assigned

**Figure 1. Spike-protein specific antibodies of classes M, G, and A in human serum.**

A  Spike-binding antibodies in serum. Binding by antibodies of classes M, G, and A is detected with secondary antibodies conjugated to different fluorochromes, and the results are shown on the left, middle, and right plots, respectively. Binding (geometric mean fluorescence intensity—GMFI—of corresponding secondary antibody) to TE cells expressing SARS-CoV-2 spike protein is plotted on the vertical axis, and binding to untransfected cells on the horizontal axis. Each point represents a value from one donor. Samples from donors with no known exposure to SARS-CoV-2 antigens (n = 86) are plotted with blue circles, recently infected donors (n = 34) with red circles, and vaccinated donors (n = 14) with green diamonds. Red circles with black triangles correspond to donors from whose B cells monoclonal antibodies were isolated (n = 5, Appendix Table S2). P values are derived from a two-way analysis of variance followed by Tukey's test to compare the specific binding, that is (binding to spike-expressing cells)/(binding to untransfected cells) between conditions (convalescent or vaccinated against unexposed) within each antibody class. The experiment was independently repeated three times, and results shown come from the third replicate.

B  Change in antibody binding following vaccination against SARS-CoV-2. Binding of antibodies from sera assessed in the experiment shown in (A) is shown for two samples of serum from each of 7 donors. Data are plotted as in (A) with binding to spike-expressing cells on the vertical axis, and to untransfected cells on the horizontal, using blue symbols for samples from before vaccination, and red symbols for samples from 15 to 29 days after the first vaccination (but before any second vaccination). Black lines connect points corresponding to pre/post pairs of samples from each donor. Left plot shows results for IgM and right plot for IgG.

C  Comparison of post-vaccination increase in binding to untransfected cells between IgG and IgM. The increase in binding to untransfected TE 0 cells between pre- and post-vaccination samples is shown on the left with blue symbols for IgG and on the right with black symbols for IgM. The horizontal line at y = 0 is the expected result when no increased binding is observed. P value is derived from a two-tailed, Wilcoxon matched pairs signed rank test.

D  Comparison of spike-specific IgM, as measured by flow cytometry or ELISA. The left two "FC" are the results of flow cytometry, and the left vertical axis shows the ratio of serum IgM binding to spike-expressing cells, vs binding to non-expressing control cells, as plotted in A. The right two columns show spike-RBD-specific IgM as measured by an IgM-capture ELISA, and the right vertical axis shows the optical densities (OD). P values are calculated by two-way analysis of variance, followed by Sidak's multiple comparisons test. The entire experiment was repeated three times, and the data shown are derived from the third replicate.

▶

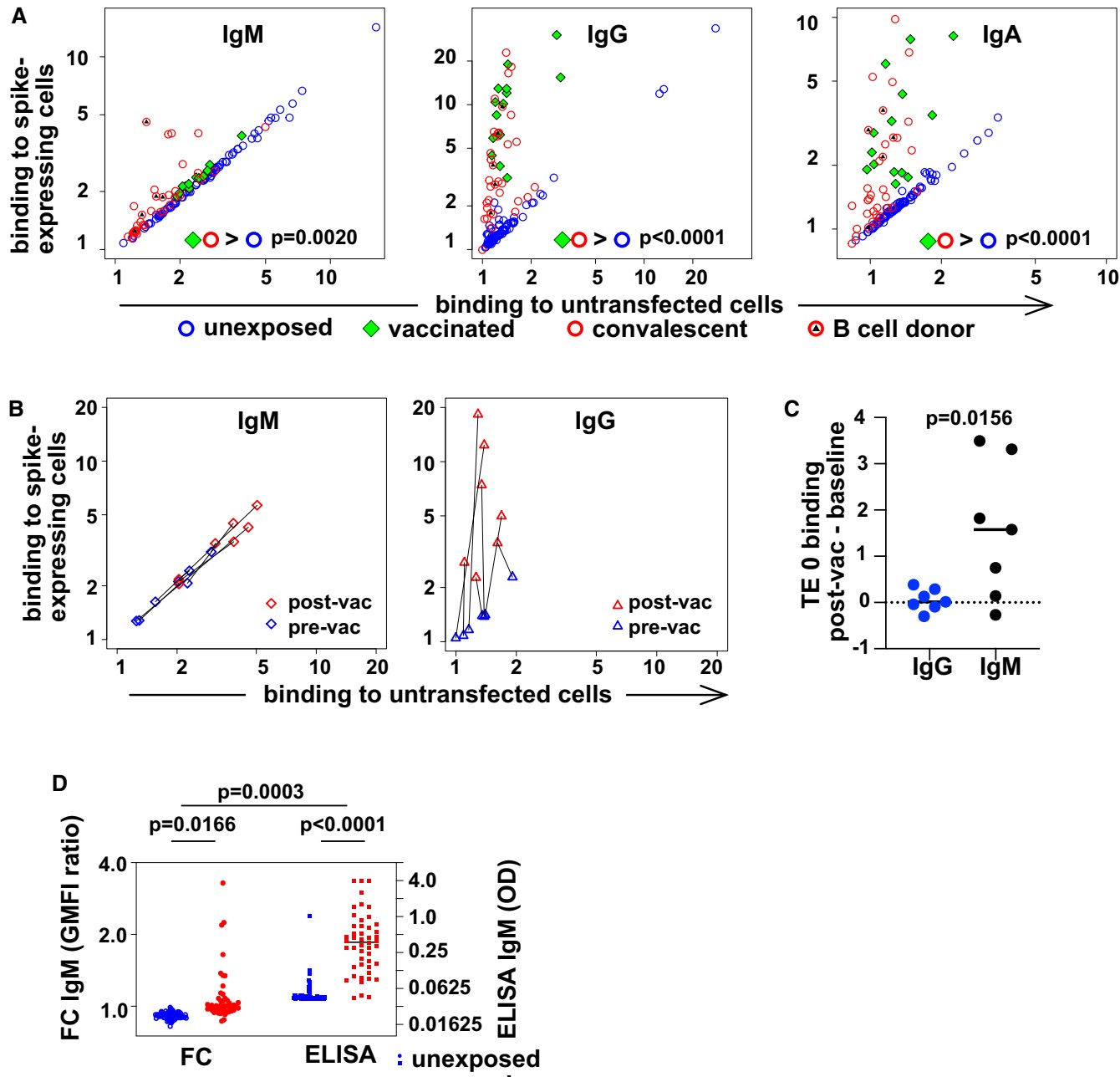

**Figure 1.**

the B cells to clusters, including memory, naive, and plasma-like (Fig 2A and B). B cells activated by capturing spike protein (Fig 2C) were predominantly memory cells of the IgM and IgG classes (Fig 2D).

To investigate the properties of the antibodies produced by these B cells, we generated monoclonal antibodies from B cells sorted with the same MACAC gate shown in Fig 2C (Fig 2E). 3266 SARS-CoV-2 spike capturing B cells from 5 randomly selected convalescent donors (infected between March and May of 2020) among those with detectable SARS-CoV-2-spike-specific antibodies in serum (Fig 1A and Appendix Table S2) were distributed into single wells of 384-well plates and cultured for 9 days in the presence of IL-21

and CD40 ligand. Resulting single cell supernatants were screened by flow cytometry (Fig 2F) and 326 (79 IgM, 141 IgG and 106 IgA) wells exhibited binding spike-specific binding activity (i.e., ratio of binding to spike-expressing cells divided by binding to control cells) above the threshold of 1.2. Unlike the polyclonal IgM in sera, these monoclonal antibody supernatants were all specific (Fig 2F). It should be noted that while the monoclonal antibodies described here are likely derived mostly from memory cells, while this may not be the case for the soluble IgM in serum.

Feldman *et al* (2021) also reported that spike-specific antibodies cloned from naïve donors exhibited no polyreactivity. One possible reason why we did not observe unspecific binding by the anti-spike

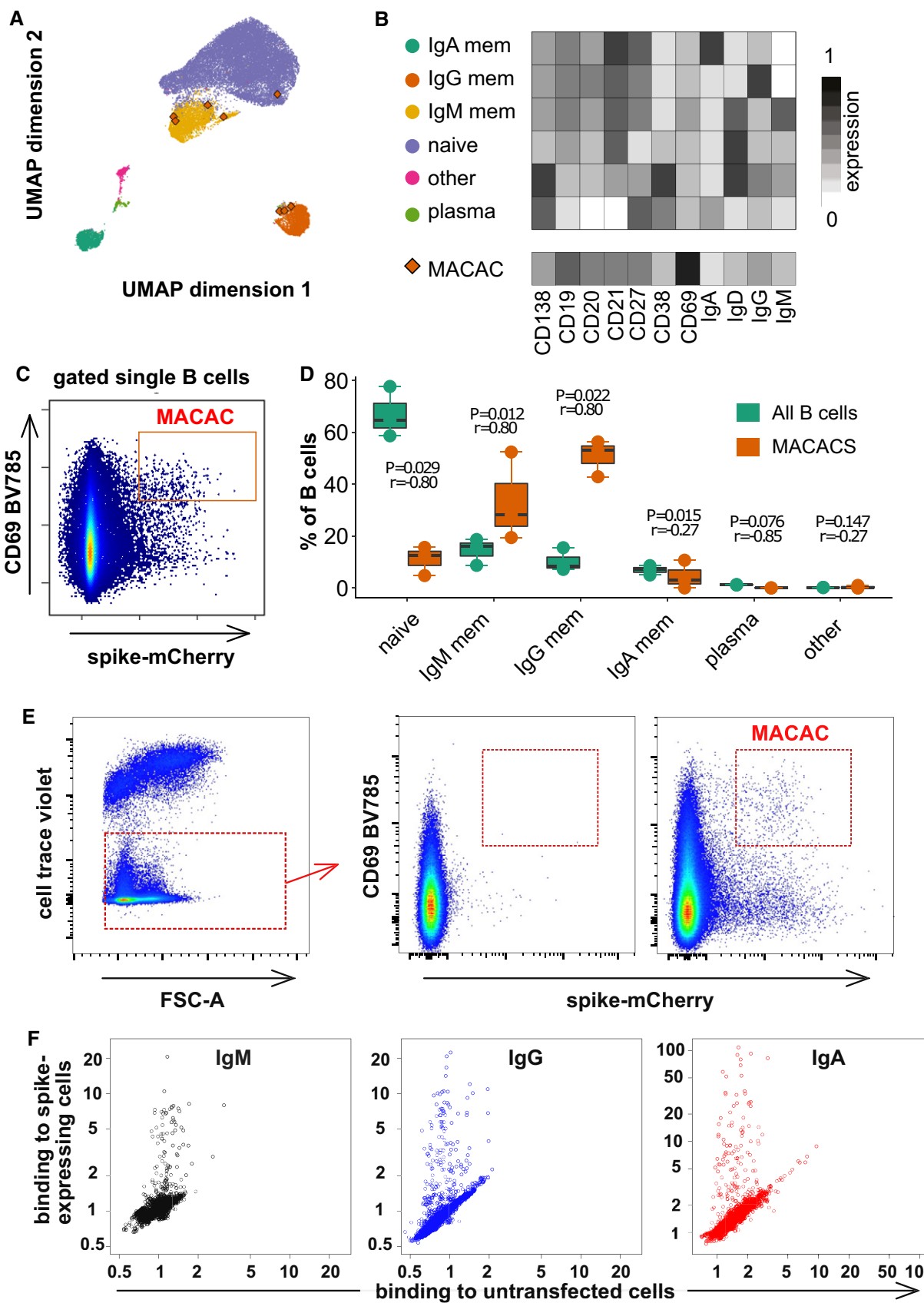

Figure 2.

**Figure 2. Spike-capturing B cells from peripheral blood of convalescent donors.**

A   Phenotypes of blood B cells. B cells from 15 ml of peripheral blood from each of three convalescent donors were isolated by negative selection with magnetic beads, exposed to adherent cells expressing spike-mCherry for 3 h, then retrieved and labeled with fluorescent antibodies shown in Appendix Table S4 and measured by flow cytometry. (A) UMAP algorithm (5,000 randomly selected cells/sample) was used to depict the major B cell subsets clustered according to marker expression. FlowSOM-based B cell subpopulations are overlaid as a color dimension, and the colors of the clusters are shown on the left of heatmap (B). Cells falling in the MACAC gate (see below) are marked with red rhombi.

B   Heatmap showing mean population expression levels of markers used for UMAP visualization and FlowSOM-clustering. Colors shown in the legend on the left are also used in the UMAP representation in (A).

C   Gating strategy to define CD69-high, mCherry-high, spike-capturing B cells (i.e., membrane antigen capture activated cells, MACAC, red box). Dot plot shows CD69 and mCherry fluorescence for B cells from one of three donors gated by forward and side scatter, and negative for the dye used to mark the antigen donor cells.

D   Relative fractions of different B cell subpopulations, from three donors, as shown in A-B, in all B cells compared to within MACAC B cells. P values are based on two-tailed t-tests between the groups. Correlation coefficients (r) were calculated from the z-statistic of the Wilcoxon–Mann–Whitney test. A black horizontal line represents the median. Boxplots represent the interquartile range (IQR). Whiskers extend to the farthest data point within a maximum of 1.5× IQR. Every point represents one donor.

E   Gating strategy in MACAC sorting. Single cells are selected based on scatter, antigen-expressing TE spike-mCherry cells excluded on Cell Trace Violet label, and the spike-capturing (mCherry-high), activated (CD69-high) B cells (population labeled "MACAC" in red on the right-most plot) are sorted. The middle plot shows B cells that did not adhere to the TE-spike-cherry antigen-expressing cells (putatively antigen-irrelevant), and the right plot those that did (putatively antigen-recognizing). Plots show data from one of five convalescent donors (those whose serum results are labeled with black triangles in Fig 1A). Cells in the MACAC gate were singly distributed into wells of 384-well plates and cultured for 9 days with IL-21 and CD40L, and then, the single well culture supernatants were screened for anti-spike antibody binding and virus neutralization as described below.

F   Results of single well supernatant screening for antibody binding to SARS-CoV-2 spike protein. Results from 3266 wells from 5 donors are shown for IgM (left), IgG (middle), and IgA (right). Screening and plotting methods are the same as in (Fig 1A).

IgM antibodies lies in the method used for isolating the B cells. Membrane antigen capture requires a higher affinity than is needed simply to bind the antigen (Natkanski et al, 2013). Recently described effective protocols for antigen-specific B cell isolation include negative selection to exclude polyspecific B cells that bind to irrelevant antigens (Robbiani et al, 2020; Thouvenel et al, 2021). MACACS achieves this by positive selection for high affinity.

**Monoclonal IgM mediate potent neutralization**

Single cell culture supernatants containing antibodies with binding activity against the SARS-CoV-2 spike protein were assessed by measuring their capacity to neutralize VSV*ΔG-S$_{\Delta21}$, a chimeric vesicular stomatitis virus (VSV) expressing the SARS-CoV-2 spike protein. Neutralization of such propagation-competent chimeric viruses has been demonstrated to be highly correlated with neutralization of SARS-CoV-2 itself, and a good readout for spike-binding (Case et al, 2020; Dieterle et al, 2020). Of the 326 supernatants we tested (Fig EV1), 31 showed detectable neutralization of the virus at a dilution of 20 or higher (11 IgM, 14 IgG, and 6 IgA). B cells from wells containing neutralizing antibodies of any of the three classes, and four non-neutralizing antibodies as controls (2 IgG, 2 IgA) were lysed, reverse-transcribed, and their transcriptomes surveyed by RNA sequencing. Immunoglobulin heavy and light genes were extracted bioinformatically then amplified from the original cDNA and recombinantly expressed in the same class as observed in the B cells for testing, yielding 6 IgA, 11 IgG, and 3 IgM. Sequences of heavy and light chains of the antibodies shown in Appendix Table S3 are available in GenBank Accession numbers OM584288-OM584289, and OM687904-OM687941 and in Dataset EV1. We observed virus-neutralizing antibodies of all three classes (Fig 3A–C).

Neutralization of the wild-type SARS-CoV-2 by the six most potent antibodies (3 IgM, 2 IgA, and 1 IgG) was assessed by determining the neutralization dose 50% (ND50), that is, the concentration of antibody at which the probability of a single cell infection event occurring is 50% (Wulff et al, 2012). ND50 values are shown in Fig 3D and range from 36 ng/ml to 679 ng/ml.

**IgM-mediated virus neutralization is not explained by complement activation**

IgM are the most effective immunoglobulin class in activating complement. It is reported that complement component C1q synergizes with IgG2 antibodies to neutralize West Nile Virus in a mouse system (Mehlhop et al, 2009). We hypothesized that the potency of the IgM antibodies observed in virus neutralization in vitro might be enhanced by the complement system. We therefore examined antigen-dependent activation of complement by anti-spike antibodies of 3 classes (Fig 3E). As expected, IgM antibodies were significantly more effective in activating complement than IgG1, and IgA antibodies induced no complement activation, despite comparable antibody binding (Fig EV2A). We next tested neutralization by IgM 2J17 in the presence of complement-sufficient human serum, or serum with the complement system degraded by heat inactivation (Fig 3F). We observed a small enhancement of neutralization in the presence complement-sufficient serum, but this difference was not statistically significant.

**Neutralization is dependent on class**

We then considered the possibility that the potency of the IgM antibodies might be a result of the higher avidity conferred by the polymeric structure. To test this, we examined the relative binding of antibodies expressed as IgM or IgG1 in a competitive setting. We prepared 3 IgM antibodies 2J17, 2E14, and 3N8 with the same light chains and variable regions of the heavy chains, but with the constant regions switched to IgG1. The artificially class-switched antibodies thus produced maintained the binding specificity of their IgM equivalents when tested individually (Fig EV2A and B). However, when the IgM and IgG were mixed and allowed to bind their target competitively, the IgM versions of each antibody out-competed their IgG partners (Fig 4A), that is, binding of IgG in presence of IgM was significantly lower than in its absence (P = 0.0101). Conversely, no effect of competition from the IgG was seen on the IgM. When we tested neutralization by these artificially class-switched antibodies, all

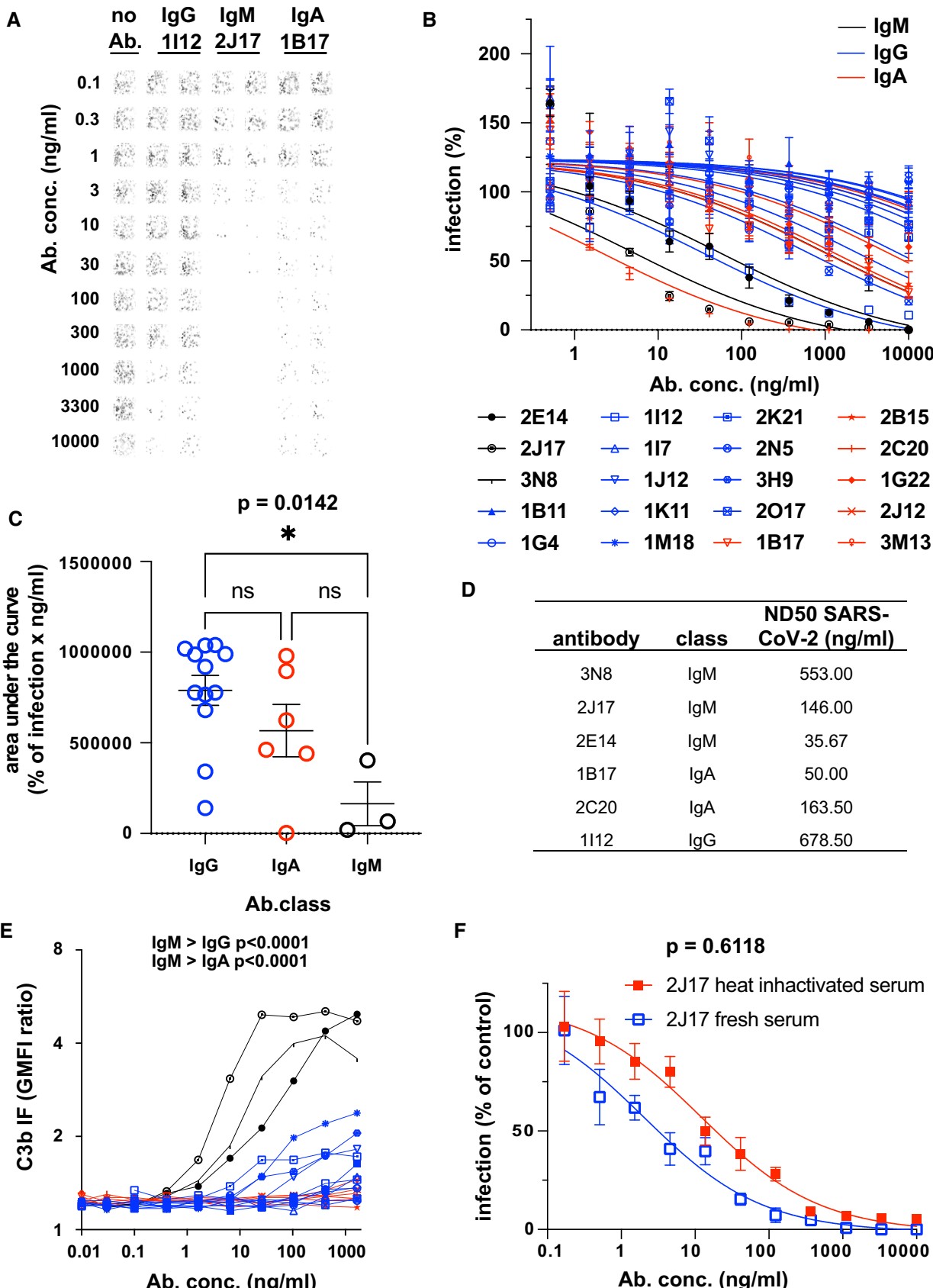

Figure 3.

**Figure 3.  Native IgM antibodies specific for SARS-CoV-2 spike protein mediate virus neutralization.**

A   Example of neutralization of chimeric VSV*ΔG-S$_{Δ21}$ by donor-derived IgG, IgM and IgA antibodies. 100 pfu VSV*ΔG-S$_{Δ21}$ was mixed with antibody at specified concentrations for 1 h and then added to Vero cells. After 24 h, the cells were imaged and infected wells identified by GFP expression, here pseudocolored with GFP-bright shown in black against a white background. Virus neutralization is manifested as reduction in GFP. Images shown here come from wells infected with virus alone (left column), virus pre-incubated with the moderately neutralizing IgG 1I12 (second and third columns), the potent neutralizing IgM 2J17 (fourth and fifth columns), or with the moderately potent IgA 1B17 (sixth and seventh columns). Each antibody was tested in the range of concentrations shown to the left of the images, from 10,000 down to 0.1 ng/ml.

B   Neutralization curves from each of the cloned antibodies using plaque reduction assay of VSV*ΔG-S$_{Δ21}$. The horizontal axis shows the serial dilution of the antibodies. The vertical axis shows, for each well exposed to antibody, the ratio of infected cell plaque count to the corresponding value from control wells without antibody. Each point is the mean of 9 values pooled from three independent experiments each with triplicate wells. Error bars indicate standard error. Antibodies tested are listed below the figure and include 3 IgM (black lines and symbols), 11 IgG (blue lines and symbols), and 6 IgA (red lines and symbols).

C   Quantification of results from (B). Area under the curve is plotted for every antibody and compared between different antibody classes by one-way ANOVA followed by Tukey's test (mean from three biological replicates and SEM). Area under the curve was chosen as a measure of neutralization capacity, rather than ND50 (as used in subsequent figures), because some antibodies included in this figure exhibited no neutralization at the concentrations tested, precluding the calculation of an ND50.

D   SARS-CoV-2 Neutralization Test. Antibodies that showed neutralization of the chimeric VSV*ΔG-S$_{Δ21}$ were tested for neutralization of wild-type SARS-CoV-2 virus. Concentrations of antibodies needed to neutralize100 pfu are expressed as ND50 (mean from two independent experiments, each with quadruplicate wells, or 3 independent experiments for 2E14 and 2J17). The ND50 of 2E14 was lower than 2J17 in three independent experiments although this difference was not significant by paired two-tailed $t$-test ($P = 0.3876$).

E   Flow cytometric determination of antibody-dependent complement deposition on cells. Cells expressing SARS-CoV-2 spike protein and untransfected control cells were incubated with various concentrations of antibody in the presence of fresh human serum (from a SARS-CoV-2 unexposed donor). Activation of the complement cascade was measured by flow cytometric assessment of complement component C3b deposition on the surface of the cells. Results for the 3 IgM, (black), 11 IgG, (blue), and 6 IgA antibodies (red) are shown. $P$ values were calculated by two-way analysis of variance, followed by Tukey's test.

F   Influence of complement on virus neutralization. A plaque reduction neutralization assay like that shown in Fig 3A–C was conducted with IgM 2J17 in the presence of either fresh, complement-sufficient human serum, or heat-inactivated serum. The concentration of 2J17 is shown on the horizontal axis. The vertical axis shows, for each well exposed to antibody, the ratio of infected cell plaque count to the corresponding value from control wells with serum (either fresh or inactivated) but without antibody. The points plotted with filled red symbols corresponds to the condition with heat-inactivated serum, and the open blue symbols results with fresh serum. Each symbol corresponds to the mean of nine wells pooled from three independent experiments, each with triplicate wells. Error bars correspond to standard error. $P$ value was calculated using a paired, two-tailed $t$-test.

three previously virus-neutralizing IgM antibodies lost neutralizing ability when expressed as IgG1 (Fig 4B and C).

If the greater neutralization potency of IgM is due to the avidity advantage of their multimeric structure, then the same advantage ought to be gained by switching a weakly neutralizing IgG or IgA to IgM. We tested virus neutralization by IgG 1I12, and 2 IgA 2C20 and 1B17, all artificially switched to IgM (Fig 4D). Results were heterogeneous, with 2 antibodies slightly increasing in potency, and one losing potency, although none of these changes was statistically significant.

Early studies of recombinantly produced human monoclonal antibodies (Tiller $et\ al$, 2008) often involved switching of IgM antibodies to IgG for recombinant production and testing, and this approach has been employed by the few workers to study the properties of IgM antibodies against SARS-CoV-2 (Feldman $et\ al$, 2021; Wang $et\ al$, 2021). Our results make clear that this exchange is likely to have a dramatic effect on their functional properties. Wang $et\ al$ (2021) successfully isolated a large number of RBD-binding monoclonal antibodies of IgM, IgG, and IgA classes; the IgG and IgA included several highly potent neutralizing antibodies, but the IgM,

**Figure 4.  Class dependency of neutralization by spike-specific IgM.**

A   The effect of competition on concentration versus binding curves of three spike-specific, class-switched antibodies. Antibodies were expressed either as IgM, as originally isolated (black symbols and lines), or artificially switched to IgG1 (blue lines and symbols). For each antibody, identified by name at the top of each plot, the binding of the antibody alone ("alone", open symbols), or the competitive binding of the antibody in a mixture of IgM and IgG1 ("compet.", filled symbols) is shown. In the competition scenario, each of the two classes of the antibody were added together at the concentration shown on the horizontal axis. The vertical axis shows the ratio of geometric mean fluorescence of human IgM, or IgG on spike-expressing cells divided by the corresponding signal on spike-non-expressing control cells. Each point shows the mean of 3 values from three independent experiments. Error bars show standard error. $P$ value was calculated by comparing the area under the curve of each antibody class in the "alone" condition with the binding of the "competition" condition by paired, two-tailed $t$-test.

B   Virus neutralization by antibodies expressed as IgM or IgG1. A plaque reduction neutralization assay as described in Fig 3A and B was used to measure neutralization by the three, donor-derived IgM (black lines and open symbols) and their artificially class-switched IgG1 equivalents (blue lines and filled symbols). Horizontal axis shows the antibody concentration and the vertical axis shows the level of infection expressed as percentage GFP expression compared to control wells with no antibody added. Each point shows the mean of nine replicate values pooled from three independent experiments, each with triplicate wells. Error bars show standard error.

C   Comparison of virus neutralization by antibodies expressed as IgM or IgG1 shown in B. Area under the curve was calculated from each of the three independently performed experiments. $P$ was calculated by paired, two-tailed $t$-test.

D   Affinities of donor-derived spike-binding IgG1 ($n = 8$) and donor-derived IgM ($n = 3$) expressed as IgG1, measured by surface plasmon resonance. $P$ value was calculated by unpaired two-tailed $t$-test. The Kd derived from each antibody measurement are shown in Appendix Table S5. Sensograms of each tested antibody are shown in Fig EV2.

E   Virus neutralization by antibodies artificially class switched from IgA or IgG1 to IgM. A plaque reduction neutralization assay as described in Fig 3A and B was used to measure neutralization by the two donor-derived IgA (pink and light blue lines) and one donor-derived IgG (Bordeaux lines). The antibodies expressed in their original classes (IgA or IgG) are plotted with filled symbols of the same colors as the lines, and their artificially class-switched IgM equivalents are plotted with open symbols. Horizontal axis shows the antibody concentration and the vertical axis shows the level of infection expressed as percentage GFP expression compared to control wells with no antibody added. Each point shows the mean of nine replicate values pooled from three independent experiments, each with triplicate wells. ND50 was calculated from each of the three independently performed experiments. $P$ was calculated by paired, two-tailed $t$-test for each antibody pair.

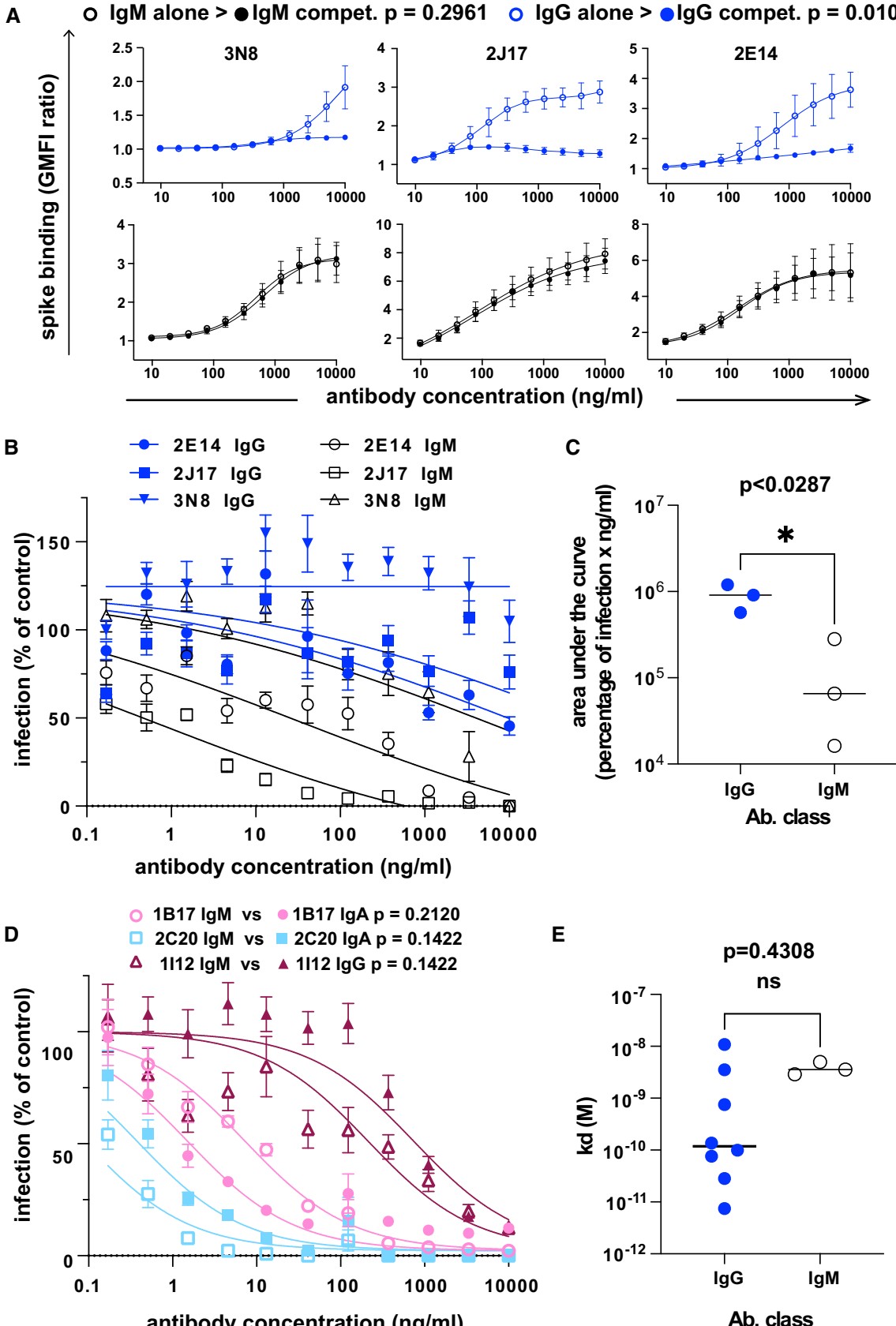

Figure 4.

which were expressed as IgG1 after artificial class switch were fewer and less potent. This is the opposite of what we observed when expressing the antibodies in their native classes. Thouvenel et al (2021) also observed that IgM antibodies against Plasmodium parasites lost their potency if converted to monomeric IgG. An obvious question is what the mechanistic basis for this IgM class-dependent potency might be. Thouvenel et al (2021) ascribe the effect of IgM to IgG class switch partly to the reduction in avidity caused by the reduction in valency, and partly to other factors, for example, enhanced steric blockade or epitope accessibility. Either conformational influence of the Fc region on the paratope (Janda et al, 2012), or the effect of Fc region flexibility on the ability of the antigen-binding domains to access the epitope (Tobita et al, 2004) might influence antigen binding.

We are not aware of any other studies of natively expressed, natural IgM monoclonal antibodies in human SARS-CoV-2, but the

phenomenon may be relevant in other contexts. Shen et al (2019) isolated two influenza-neutralizing antibodies from influenza virus-infected mice 7G6-IgM and 3G10-IgM. When made into a chimeric antibody with human IgG1 constant regions, the potency of the more potent antibody 7G6-IgM fell by around 100-fold, while the less potent 3G10-IgM did not change significantly. The IgM version also offered better *in vivo* protection against challenge with various influenza strains than did the IgG version.

### Affinity of IgM variable regions is at lower end of the range shown by IgG

To directly measure binding affinities of antibodies of different classes, we used surface plasmon resonance (SPR) to measure the affinities of the 3 IgM (2E14, 2J17, 3N8) and 8 antibodies originally isolated as IgG. To eliminate the confound of valency, all 11

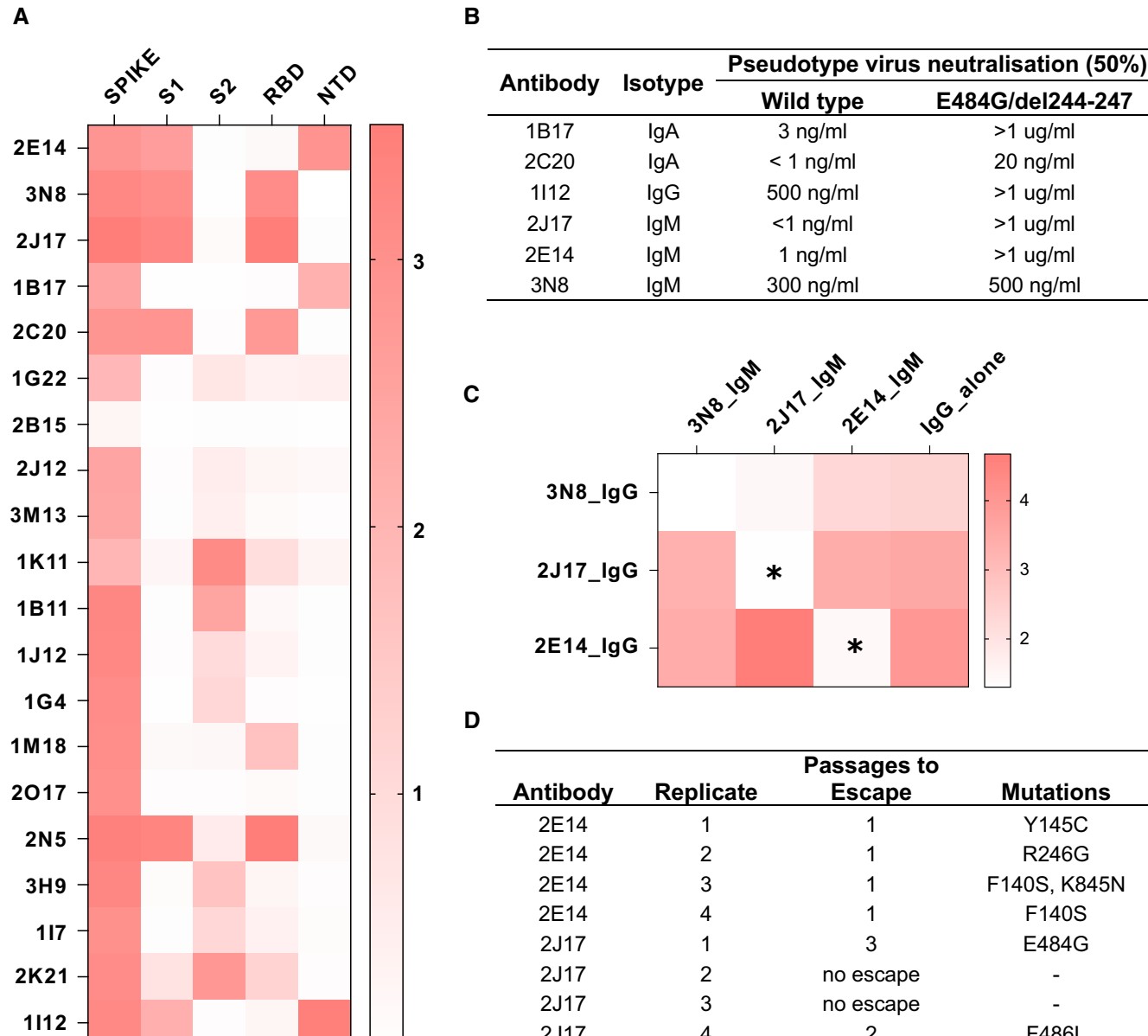

**Figure 5.**

**Figure 5. Epitopes recognized by spike-binding antibodies.**

A   Heatmap of antibody binding to spike protein subdomains (complete spike extracellular domain "spike", S1, S2, RBD and NTD in ELISA. Color gradient shown on right represents optical density at 450 nm (OD). Values are mean OD from two independent experiments. Analogous ELISA results for 3 IgM and their IgG1-class-switched derivatives are shown in Fig EV2B.

B   Susceptibility of different neutralizing antibodies to spike protein mutations. A neutralization assay was performed with 3 IgM, 1 IgG and 2 IgA using VSV*ΔG(FLuc) pseudotyped with wild-type (wt) spike protein or with spike variant harboring the deletion of amino acids 244–247, located in the NTD, and the substitution E484G, located in the RBD, from a SARS-CoV-2 isolate from an immunosuppressed COVID-19 patient (del244-247/E484G). ND50 values were determined from quadruplicate wells, from an experiment performed once. Neutralization curves are shown in Fig EV4.

C   Flow-cytometric epitope binning of monoclonal IgM, utilizing competitive displacement of IgG by IgM with identical variable region (as shown in Fig 4A). Cells expressing SARS-CoV-2 spike protein and untransfected control cells were incubated with each of 3 neutralizing IgM antibodies artificially switched to IgG1, either alone, or mixed with one of the 3 antibodies expressed as IgM. The binding of each IgG1 alone is shown in the last column of the heatmap, and in competition with each other IgM in columns 1-3. Color shade represents the specific IgG binding (ratio of GMFI on TE spike-mCherry cells to TE 0 cells). Values are mean GMFI ratios from three independent experiments. When IgM binds the same epitope as the IgG (automatically the case when the source antibody is the same), the IgM will displace the IgG, resulting in a reduction in the IgG signal. Asterisks mark those combinations of antibodies with a statistically significant decrease in binding of the IgG1 in the presence of IgM, compared to the IgG alone across 3 independent experiments, (*$P < 0.05$, two-way analysis of variance, followed by Tukey's test).

D   Summary of escape mutations induced by potent neutralizing IgM 2E14 and 2J17. Vero cells in 24-well plates were infected with VSV*ΔG-S$_{Δ21}$, in the presence of serial dilutions from 10 µg per ml of neutralizing antibody (or without antibody as control), and 2 days later, virus from the well containing the highest concentration of antibody that still showed some virus proliferation was used to infect fresh wells of Vero cells, again in the presence of antibody in a serial dilution. The process was repeated for three passages, and at the last passage, the cells were lysed, RNA extracted, reversed transcribed and sequenced, and consensus spike gene reads extracted and aligned against the reference SARS-CoV-2 genome MN908947.3. The table shows de novo amino acid mutations that arose in each of four replicates with each of the two neutralizing antibodies, as well as the number of passages needed for the escape mutant to emerge. In two of the replicates with 2J17, no escape was observed. Consensus sequences of spike genes from each replicates and from control wells with no antibody added are shown in Dataset EV2.

antibodies, regardless of original class, were expressed as IgG1. We initially used whole spike protein as target antigen, but some antibodies failed to bind this antigen in the SPR apparatus, and for these, we used the subdomain of the protein bound by that antibody (see results below on epitope specificity). Antibodies originally isolated as IgG had a wide range of $K_d$ from 28 pM to 11 nM. When expressed as monomeric IgG1, affinities of the antibodies originally isolated as IgM were at the lower end of this range, with the highest affinity for 2E14 at $K_d = 2.89$ nM (Fig 4E, Appendix Table S5 and Fig EV3). The one neutralizing IgG that we isolated, 1I12, had an affinity below that of the IgM-derived antibodies, suggesting that it is not because of inadequate monomeric affinity that the IgM antibodies fail to neutralize when switched to IgG.

The monomeric (i.e., bivalent) affinity of the recombinant antibodies is relevant to the question regarding the antibodies described here is whether they are representative of those that are secreted *in vivo*. To obtain T cell help, necessary for a high affinity antibody response, a B cell must not only bind an antigen, but internalize it for processing and presentation. The dependence of this mechanism on antigen affinity was well established by Batista and Neuberger (1998), Batista and Neuberger (1998). Although an IgM antibody can achieve enhanced avidity by virtue of higher valency, this effect is thought not to be relevant to B cell activation, because the monomeric affinity of the BCR is thought to be the parameter that determines BCR signaling and antigen uptake (Natkanski *et al*, 2013). We propose that an important advantage of the MACACS technique is the specific identification of those B cells whose monomeric BCR affinity for the target antigen is high enough to enable BCR signaling, antigen uptake, and presentation, and thereby the acquisition of T cell help and differentiation into plasma cells. The fact that the neutralizing IgM we examined would have been functionally impaired if secreted as IgG raises the question of how a B cell avoids premature class switch. The results shown here suggest that a B cell ought not to switch from IgM to IgG until it has achieved a certain affinity through maturation, but do not explain how this decision is made. One attractive idea is that the soluble, high avidity IgM might out-compete IgG of a similar monomeric affinity, thereby forcing the

evolution of B cells with higher affinity BCR, as suggested by Zhang *et al* (2013). The opposite perspective is that beneficial antibodies can be produced in the IgM class even by B cells whose monomeric BCR affinity is modest. By this logic, to make the decision whether to become activated in response to an encountered multivalent antigen, a B cell ought to probe the antigen with multiple BCR, and integrate the signals. This may be one function of the spreading and contraction response of B cells (Fleire *et al*, 2006) on which the MACACS technique is based. Such multi-BCR antigen scanning would offer an alternative mechanism for class-appropriate affinity testing, if the signal required for plasma cell differentiation and antibody secretion were to have a higher threshold for class-switched B cells than for IgM B cells.

The results described here do not enable us to determine the mechanistic basis for the class-dependent potency of 2J17 and 2E14. Competitive displacement of IgG by IgM with the same variable region is consistent with the widely postulated higher avidity of IgM, but since the measured affinities of the IgG1 derivatives of the neutralizing IgM have affinities similar to the neutralizing IgG 1I12, the reduction of avidity caused by class switch cannot be the only factor. Similarly, the effects of Fc region on the paratope, and epitope accessibility are hypothesized to affect binding, and therefore by the same logic are insufficient to explain the loss of neutralization. Yet another possibility is a post-entry mechanism involving signaling of the attached antibody to the infected cell (Green *et al*, 1992).

## Neutralizing IgM target RBD and NTD

Precise binding specificity to the spike protein was investigated by ELISA using recombinant subdomains as antigen. Binding of each of the 3 IgM, 6 IgA, and 11 IgG to the whole spike extracellular domain, the S1 and S2 subdomains, the N-terminal domain (NTD) and the receptor-binding domain (RBD) are shown as a heatmap in Fig 5A. One IgA (2B15), despite showing replicable binding to spike-expressing cells, showed no binding to any of these antigens in ELISA, an observation also made by Wang *et al* (2021) (Wan

et al, 2020). All antibodies identified as neutralizing recognized either the RBD or the NTD. We examined the impact of mutations in these domains by measuring neutralization of a virus pseudo-typed with the spike protein of a clinical SARS-CoV-2 isolate with mutations in NTD (del244-247), and the RBD (E484G) (Figs 5B and EV4) (Weigang et al, 2021). Neutralizing potency of the RBD-binding IgM 2J17 and the NTD-binding IgM 2E14 both fell to below the detection limit of ND50 > 1 µg/ml when confronted with the mutated spike protein. The effect of the mutations on the RBD-binding IgA 2C20 was much more modest, retaining an ND50 of 20 ng/ml. We compared the epitope specificities of the IgMs with one another by a flow cytometric method exploiting the effect shown in Fig 4A, that IgM can competitively displace their IgG equivalents with the same variable domain (Fig 5C). Spike-expressing cells labeled with mixtures of IgM and IgG showed reduced binding of IgG in the presence of IgM of the same specificity. IgM 2E14 had no effect on binding of either 2J17 or 3N8 confirming that 2E14 binds an independent epitope. The potency of 2E14 and 2J17 enabled investigation of their fine epitope specificity by escape mutation studies. We passaged the chimeric virus VSV*ΔG-S$_{\Delta 21}$ in the presence of each of the neutralizing antibodies until escape was observed and then sequenced the spike genes (Fig 5D, Dataset EV2). IgM 2E14 induced mutations in the NTD at residues F140, Y145, and R246. 2J17 suppressed escape in some replicates completely, but where escape occurred, the mutations were located in the RBD at residues E484 and F486.

In terms of their epitope specificity, 2J17 and 2E14 seem to function like numerous reported neutralizing IgG, by binding the region around E484 in the RBD, and the supersite in the NTD, respectively. Mutations of E484 are well known to mediate escape from neutralizing antibodies and are present in the beta and gamma variants of the virus (Harvey et al, 2021). The mutation E484K is selected by the neutralizing antibody C121, which uses the same VH gene 1-2 used by 2J17 (Robbiani et al, 2020; Weisblum et al, 2020). The combination of mutations at F140 and R246 is characteristic of escape from antibodies binding the NTD antigenic supersite described by McCallum et al (2021). The structures of the antibodies are not obviously typical of reported anti-spike antibodies; neither of them uses a VH gene identified as over-represented in RBD-binding antibodies by Robbiani et al (2020), although the KV1-33 used by 2J17 was also over-represented in this sample. The CoV-AbDab database (Raybould et al, 2021) of SARS-CoV-2-related antibodies includes two examples of antibodies with the same combination of VH and VL genes used by 2J17 and 2E14, namely C004 described by Robbiani et al (2020), and COVA2-26 described by Brouwer et al (2020), respectively. The antibodies in the CoV-AbDab database with the highest degree of CDRH3 similarity to 2J17 and 2E14 have only 53% and 25% identity, respectively, offering no evidence that these two antibodies are part of an identified public clonotype.

Our study makes clear that the influence of isotype must be considered when investigating the properties of naturally produced antibodies. The powerful influence of class on neutralization we observed here was unexpected, but other important functions of antibodies are known to be class- and subclass-dependent. The influence of antibody class on complement activation, for example, is well established (Lu et al, 2018). Antibodies of different classes also affect the immune system via class-specific Fc receptors in different ways (Boudreau & Alter, 2019; Zohar et al, 2020). Using

membrane antigen capture to identify high affinity antigen-specific B cells reveals properties of IgM in acute infection that may previously have been obscured by abundant polyreactive B cells and antibodies. The next challenge will be to develop assays that visualize the complex, class- and specificity-dependent interactions of antibodies with the rest of the immune system, that are nonetheless simple enough to be used at revealingly high throughput.

# Materials and Methods

## Blood samples

Serum and blood samples were donated by consenting convalescent, vaccinated, or unexposed donors. A subset of the donors, including all those from whom B cells were taken for monoclonal antibody isolation, were recruited as part of a trial of convalescent plasma donation as a therapy for COVID-19. Vaccinated donors all received the Moderna COVID-19 mRNA-1273 vaccine, and donors designated "fully vaccinated" received two doses. The project was reviewed and authorized by the Ethikkommission Nordwest und Zentralschweiz, Project Number 2021-00961. Blood for cell isolation was collected in S-Monovette tubes (S-Monovette® K3 EDTA, 7.5ml, REF 01.1605.100) with EDTA. Blood for serum was collected in S-Monovette tubes (S-Monovette® Serum, 7.5 ml, REF 01.1601.100) and allowed to coagulate for 1 h at room temperature. Serum was separated from coagulated blood by centrifugation at 2,000 g at 20°C, and stored at −20°C until use. Peripheral blood mononuclear cells (PBMC) were separated from EDTA-treated blood by density gradient centrifugation over Lymphoprep (Axon Lab, D015644) according to the manufacturer's instructions, mixed with 90% fetal calf serum (FCS) and 10% dimethyl sulfoxide, frozen in an isopropanol-containing freezing box at −80°C and then stored in or over liquid nitrogen until use.

## Biosafety

Experiments with pseudotyped, vesicular stomatitis-derived virions were conducted in the University of Basel at Biosafety level 2. Experiments involving live SARS-CoV-2 were conducted at the Institute for Virology and Immunology in Mittelhäusern, Switzerland, at BSL3, under permits from the Bundesamt für Gesundheit (Swiss Federal Health Department).

## SARS-CoV-2 spike protein and cell lines

A sequence-optimized DNA fragment encoding the SARS-CoV-2 spike protein (uniprot accession number P0DTC2) was cloned into the pcDNA6 plasmid between HindIII and EcoRI sites. This fragment was amplified by PCR and fused in frame with mCherry by cloning into pcDNA3 mCherry LIC cloning vector (a gift from Scott Gradia, Addgene plasmid # 30125). This plasmid was used to prepare the cell line TE spike-cherry by transfection with jetPrime (Polyplus, PPLU114-15), and selection with 0.5 µg/ml puromycin. pCMV3 encoding human CD40 ligand was purchased from Sino Biological (HG10239-UT). Transfected TE 671 cells were grown under hygromycin selection, sorted for CD40L expression using PE-conjugated mouse anti-human CD154 (BD Pharmingen, 557299), irradiated at 72 Gy for mitotic inactivation, and kept frozen in liquid nitrogen

until use. Vero cells were a gift from Daniel Pinschewer. TE 671 cells were purchased from ATCC (LGC, Wesel, Germany, CRL-8805) and were cultured following the manufacturer indications. Suspension serum-free FreeStyleTM293-F cells (Life Technologies) cultured in FreeStyleTM293 Expression Medium (Life Technologies) were grown in 125 ml sterile Erlenmeyer flasks with vented caps at densities between 300,000 and 500,000 viable cells/ml, rotating at 135 rpm on an orbital shaker platform. Cells were tested for myco-plasma contamination yearly.

### SARS-CoV-2 IgM-capture ELISA

Serum IgM reactivity against SARS-CoV-2 was assessed using WANTAI SARS-CoV-2 IgM ELISA (WS-1196) according to the manu-facturer's instructions. Briefly, microwell strips pre-coated with anti-body directed against human IgM were incubated with 1:10 diluted human serum, washed five times, then incubated with HRP-conjugated SARS-CoV-2 spike protein RBD antigen, and developed after five washes using the chromogen solution. The reaction was stopped with sulfuric acid, and plates were read at 450 nm immedi-ately after stopping.

### Monoclonal antibody ELISA

384-well plates were coated with goat anti-human IgG (Southern Biotec, 2014-01), anti-human IgM (Southern Biotec, 2023-01), or anti-human IgA (Southern Biotec, 2053-01) antibodies overnight at 4°C, then washed once with PBS and blocked with PBS- 1% BSA at room temperature for 90 min. Plates were then washed three times with PBS 0.05% Tween, incubated with 15 µl of serially diluted samples for 2 h at room temperature, washed three times with PBS 0.05% Tween, and incubated with anti-IgG-HRP (Southern Biotec 2014-05), anti-IgM-HRP (Southern Biotec 2023-05), anti-IgA-HRP (Southern Biotec 2053-05), or goat anti-Ig-HRP (Southern Biotec, 2010-05) in PBS-0.1% BSA for 1h at room temperature. Plates were then washed three times with 80 µl/well of PBS-0.05% Tween and developed with TMB ELISA substrate (SureBlue Reserve TMB Microwell Peroxidase Substrate, REF 53-00-00) until a blue color was visible; the reaction was stopped with sulfuric acid and plates were read at 450 nm immediately after stopping.

### Assessment of spike subunit specificity by ELISA

Spike subunit ELISA was based on the method described by Rodda et al (2021). 96-well plates (Corning) were coated with 2 µg/ml of recombinant SARS-CoV-2 S1+S2 (Sino Biological 40589-V08H4), SARS-CoV-2 S1 (Sino Biological 40591-V08H), SARS-CoV-2 S2 (Sino Biological 40590-V08H1), SARS-CoV-2 RBD (Sino Biological 40592-V08H), or SARS-CoV-2 NTD (Sino Biological 40591-V49H) diluted in PBS and incubated at 4°C overnight. Plates were washed with PBS-T (PBS containing 0.05% Tween-20) and blocked in 2% BSA for 1h at 37°C. Monoclonal antibodies diluted at 10 µg/ml in 0.1% BSA were added and incubated at 37°C for 1 h. After washing three times in PBS-T, secondary antibodies diluted in 0.1%BSA were added to the wells. Secondary antibodies used were: anti-human IgG-HRP (Southern Biotec 2014-05), anti-IgM-HRP (Southern Biotec 2023-05), anti-IgA-HRP (Southern Biotec 2053-05). Plates were incubated with secondary antibodies for 1 h at 37°C. Plates were then washed three

times with PBS-T and developed with TMB ELISA substrate (Sure-Blue Reserve TMB Microwell Peroxidase Substrate, REF 53-00-00) until a blue color was visible; the reaction was stopped with sulfuric acid and plates were read at 450 nm immediately after stopping.

### Spike-specific B cell isolation by MACACS

MACACS was conducted as described by Zimmermann et al (2019), adapted to isolated B cells specific for the spike protein of the SARS-CoV-2 virus. PBMC were thawed and rested for 1 h in RPMI-10 at 37°C, and then, B cells were isolated from with the Pan B cell isolation, human kit (Miltenyi, 130-101-638). B cells were then added to an adherent layer of TE spike-cherry that had been prelabelled with 5 µM of Cell Trace Violet (Invitrogen, C34557). After 20 min, non-adherent B cells were washed away gently, and the cells adhering to the cell layer were incubated at 37°C for a further 160 min. B cells were then vigorously washed off with DPBS, collected by centrifugation, labeled with Brilliant Violet 785-conjugated anti-CD69 antibody (Biolegend 310932) and sorted in a flow cytometric cell sorter (BD SORPAria III), using the gating strategy shown in Fig 1D. Sorted mCherry-high, CD69-high B cells were cultured in 384-well plate wells at an average density of 0.9 cells/well, in RPMI with 40% FCS, 0.05 ng/µl IL-21 (Gibco, PHC0215), together with 50,000 irradiated TE CD40L cells/well. After 9 days of culture, 15 µl of supernatant was withdrawn from each well and screened for spike-binding antibodies as described above. B cells from positive wells were lysed in 20 µl of 15 mM Tris–HCl, pH 8.0 containing 0.5 U/µl of recombinant murine RNAse inhibitor (NEB, M0314L). RNA was extracted from this lysate by Zymo Quick-RNA microprep kit (R1050 & R1051), and reverse transcribed with the SMART-Seq v4 Ultra Low Input RNA Kit for Sequencing (Takara, 634888). One hundred nanograms of the thus generated cDNA was used to prepare sequencing libraries with the DNA prep kit (Illumina). The libraries were sequenced paired-end 150 bp on a NextSeq 500 sequencer (Illumina) yielding an average of one million reads per lysate. Reads mapping to immunoglobulin gene sequences were extracted from the resulting single-cell transcriptomes with a custom script in R (Dataset EV3 and 4), using packages Shortread (Morgan et al, 2009) and Biostrings, reassembled with EZassembler (Masoudi-Nejad et al, 2006), and assigned to V(D)J genes using IgBLAST (https://www.ncbi.nlm.nih.gov/igblast/).

Flow cytometric characterization of the mCherry-capturing and CD69-high MACAC population was based on the MACACS proce-dure with three modifications. Non-adherent B cells were not removed after 20 min; in addition to the anti-CD69 antibody, cells were labeled with the cocktail of fluorescently labeled antibodies specified in Appendix Table S4 and instead of sorting, fluorescent signals were analyzed on an LSRFortessa cytometer (BD Bios-ciences). Single B cells were gated and transformed as previously described (Galli et al, 2019; Diebold et al, 2022). Briefly, data were transformed using the arcsinh function of the R environment. Cofac-tors were automated determined using the FlowVS package. Percen-tile normalization was then applied ranging from 0.01th and 0.999th percentile. Clustering and meta-clustering were performed using the FlowSOM package, using following parameters: IgA, IgG, CD38, CD27, CD20, IgM, CD21, CD138, IgD. A UMAP was built using the same parameters (n neighbors = 50, min epochs = 200, min

dist = 0.1). MACACS cells were defined as B cells expressing CD69 > 0.7 and mCherry > 0.5. All plots were drawn using ggplot2.

## Cloning and production of recombinant antibodies

Based on the deduced heavy and light chain sequences, two pairs of primers were designed to target the 5′ untranslated regions of the V gene, and the 3′ untranslated region of the constant regions (Appendix Table S6). The heavy and light chains were amplified from the same single-cell lysate-derived cDNA used for sequencing by one round of PCR using Phusion polymerase (NEB, M0530S) with 25 cycles, following the manufacturer's instructions. Light and heavy chain amplicons were cloned into the MCS1 and MCS2 sites of the pVITRO hygro dual expression plasmid (Invivogen, pvitro1-mcs), and the resulting plasmids transfected into Freeestyle HEK cells (Invitrogen, R79007). On the day of transfection, each 125-ml flask containing 14 ml of cells at $1 \times 10^6$ viable cells/ml was transfected with pVITRO hygro dual expression plasmid expressing the heavy and the light chain of each of the recombinant antibodies, using 293-fectin Transfection Reagent (Gibco, 12347019) according to the manufacturer's instructions. 24 h post-transfection, hygromycin B (50 µg/ml) was added to transfected cells, which were then maintained in culture under selection for 2 weeks at densities between $3 \times 10^5$ and $5 \times 10^5$ viable cells/ml. Cultured supernatants were harvested after 16 days, centrifuged at 2,000 g for 20 min, passed over 0.45 mm filters (Sartorius). Cell supernatants were buffer-exchanged into PBS using Amicon Ultra 15 50-K columns (Sigma, UFC905024) following the manufacturer's instructions, and the resulting antibody-containing supernatants were stored at 4°C or at −20°C until use. Recombinant antibodies were quantified by ELISA.

## In vitro class switch

To switch the immunoglobulin expression plasmids from mu to gamma, we fused the variable region of the heavy chain to the gamma 1 constant region by a two-step fusion PCR with the primers shown in Appendix Table S7. In the first step, we amplified the variable region of the heavy chain from the pVITRO expression plasmid with a forward primer in the plasmid backbone (pVITRO_MCS2_for_out) and a reverse primer whose first 24 bases are complementary to the 5′ end of the gamma 1 constant region, and whose last 19 bases are complementary to the 3′ end of the J gene (J gene specific, e.g., IGHJ4_Gcon_fus_rev). At the same time, we amplified the gamma 1 constant region from a pVITRO expression plasmid encoding an IgG1 antibody, using the primer pair (gamma1_constant_for and pVITRO_MCS2_rev_out). In the second step, we fused these two products, using 10 ng of each purified first step product as template, a forward primer matching the pVITRO backbone just 5′ to the MCS2 cloning site (pVITRO_MCS2_for_in), and a reverse primer complementary to the pVITRO backbone, just 3′ to the MCS2 cloning site (pVITRO_MCS2_rev_in). The fused product was cleaned with Macherey Nagel gel and PCR cleanup kit (740609) according to the manufacturer's instructions, cut with EcoRV and NheI and cloned into the original IgM encoding plasmid after excision of the heavy chain coding region with the same two restriction enzymes. The resulting plasmid was transfected into Freestyle cells as described above. Class switch from IgG or IgA to IgM was achieved by an exactly analogous procedure, using

the primers J3_02_rev_fus_to_mu or J6_02_rev_fus_to_mu, depending on the J gene of the source heavy chain.

## Antibody purification

Antibody-containing cell culture supernatants were buffer exchanged into PBS using Amicon Ultra 15 50k columns (Sigma), to a final volume of 10 ml. For IgM purification, ammonium sulfate (Sigma-Aldrich A4418) was added to IgM-containing supernatants to reach final concentration of 0.7 M. Solid ammonium sulfate was supplemented to the sample in small amounts while continuously stirring to avoid precipitation of IgM and the sample was passed through a 0.45 µm filter.

Column operations were performed at constant flow rate of 1 ml/min. HiTrap IgM Purification HP column (Sigma-Aldrich GE17-5110-01) was equilibrated with 10 ml binding buffer (20 mM sodium phosphate pH 7.5, 0.7 M ammonium sulfate). Supernatant was applied to the column twice and the column was washed with 15 ml binding buffer. Elution of bound IgM was performed with 20 mM sodium phosphate pH 7.5 using a one-step gradient and 0.5 ml elution fractions were collected. 20 µl from each elution fraction were analyzed on Superdex® 200 Increase 10/300 GL SEC column (Sigma-Aldrich GE28-9909-44) connected to a LC-4000 Series HPLC System (JASCO). Sample-containing fractions were pooled and concentrated, and IgM were purified with a final run of Superdex 200 Increase.

For IgA purification, column operations were performed at constant flowrate of 1 ml/min. HiTrap KappaSelect (Sigma-Aldrich GE17-5458-11) and HiTrap LambdaFabSelect (Sigma-Aldrich GE17-5482-11) columns were equilibrated with 10 ml phosphate-buffered saline (PBS, 10 mM phosphate buffer pH 7.4, 2.7 mM KCl, 140 mM NaCl). The IgA-containing supernatants were applied to the column (KappaSelect for 3M13, 1G22, 2J12, 1B17, 2B15 and LambdaFabSelect for 2C20) twice and the column was washed with 15 ml PBS. Elution of bound IgA was performed with 0.1 M glycine buffer, pH 2.7 (from KappaSelect) or 0.1 M acetate buffer, pH 3.5 (from LambdaFabSelect) using a one-step gradient and 0.5 ml elution fractions were collected. Eluted fractions were immediately adjusted to physiologic pH with the addition of 50 µl 1 M Tris–HCl pH 8.0.

IgG were purified using Protein G HP SpinTrap (Cytiva 28903134) following the manufacturer's instructions.

## Neutralization assays

Three different assays were used to assess virus neutralization by antibodies. Supernatants from expanded single B cells were tested using GFP-expressing vesicular stomatitis virus pseudotyped with SARS-CoV-2 spike protein (chimeric GFP-expressing vesicular stomatitis virus). Monoclonal antibodies were tested with this same assay, and a subset was also against wild-type SARS-CoV-2 by a classical plaque assay; and the impact of a clinically relevant mutation in the spike protein on neutralization was assessed using a luciferase-expressing VSV (VSV*ΔG(FLuc) pseudotypes) pseudotyped with either mutant or wild-type spike protein, using a luciferase endpoint instead of GFP plaque measurement. Each neutralization assay was performed either two or three times, as indicated in the corresponding figure legend, on three separate days,

using the same antibody preparation; each experiment included triplicate or quadruplicate wells for every antibody dilution step.

## Plaque reduction assay using chimeric GFP-expressing vesicular stomatitis virus

A modified SARS-CoV-2 spike cDNA was synthesized by GenScript (Leiden, The Netherlands). The cDNA was based on the sequence of the Wuhan-Hu-1-Isolate (NC_045512.2) but the encoded spike protein had the cytoplasmic domain truncated by 21 amino acids. The S1/ S2 cleavage site contained the mutation R685G. In addition, the spike protein harbored several mutations which are likely a consequence of the adaptation to VeroE6 cells: H655Y, D253N, W64R, G261R, A372T (Dieterle et al, 2020). The cDNA was cloned into the pVSV*ΔG(MERS-S) plasmid vector to replace the MERS-CoV- spike gene and the chimeric virus VSV*ΔG-S$_{\Delta21}$ was generated as described previously (Pfaender et al, 2020).

For the plaque reduction assay, Vero cells were seeded at 3000 cells per well into 384-well plates ("assay plates") in Dulbecco's modified Eagle's medium (DMEM) with 10% fetal calf serum, 100 U/ml penicillin, and 100 μg/ml streptomycin added (DMEM-10) and grown overnight. The next day, antibodies were mixed at various concentrations in 384-well plate wells ("mixing plates"), each with 100 plaque forming units of VSV*ΔG-S$_{\Delta21}$ in 40 μl DMEM, incubated for 1 h at 37°C, then added to Vero cells in the assay plates. After one 90 min, 40 μl of DMEM with 1% methylcellulose was added to each well, and the assay plates incubated for a further 24 h at 37°C. Plates were then imaged in the GFP channel with an automated fluorescent microscope (Nikon Ti2), and data saved as.nd2 files. Numbers of GFP-expressing (i.e., infected) plaques were extracted using an automated macro in ImageJ. For comparison of neutralization between conditions, ND50 (calculated in Prism using four parameter non-linear regression) was used where possible; when this was not possible because one or more of the conditions showed no detectable neutralization, the area under the infection/concentration curve was used instead.

## SARS-CoV-2 Neutralization by classical plaque assay

Serial dilutions of control sera and samples were prepared in quadruplicates in 96-well cell culture plates using DMEM cell culture medium (50 μl/well). To each well, 50 μl of DMEM containing 100 tissue culture infectious dose 50% (TCID50) of SARS-CoV-2 (SARS-CoV-2/München-1.1/2020/929) were added and incubated for 60 min at 37°C. Subsequently, 100 μl of Vero E6 cell suspension (100,000 cells/ml in DMEM with 10% FBS) were added to each well and incubated for 72 h at 37°C. The cells were fixed for 1 h at room temperature with 4% buffered formalin solution containing 1% crystal violet (Merck, Darmstadt, Germany). Finally, the microtiter plates were rinsed with deionized water and immune serum-mediated protection from cytopathic effect was visually assessed. Neutralization doses 50% (ND50) values were calculated according to the Spearman and Kärber method.

## Neutralization of VSV*ΔG(Fluc) pseudotypes

The spike protein cDNA of SARS-CoV-2 containing the D614G mutation and a truncation of the cytoplasmic domain by 18 amino acids but otherwise based on based on the Wuhan-Hu-1 reference sequence (NC_045512.2) was cloned into the pCAGGS plasmid taking advantage of the KpnI and XhoI endonuclease restriction sites. A spike variant containing the E484G substitution and a deletion of amino acids 244–247 were generated by overlapping PCR technology. Transcomplementation of the VSV*ΔG(Fluc) vector (Rentsch & Zimmer, 2011) was performed as described recently (Zettl et al, 2020). Antibody-mediated neutralization of the pseudotyped VSV*ΔG(Fluc) was performed by taking advantage of the firefly luciferase reporter as described previously (Zettl et al, 2020).

## Flow cytometry

For flow cytometry screening of spike-binding antibodies in serum, 100,000 TE 0 cells and the same number of TE spike-cherry cells per well were incubated with serial 2-fold dilutions of serum, starting from 1:20 in PBS. Incubation was done in 96-well plates, for 30 min, on ice. Cells were then washed twice with cold PBS and labeled with 100 μl of PBS containing Dylight-405-conjugated anti-human IgM (JIR 109-475-129), FITC-conjugated anti-human IgG (JIR 109-096-098), and Alexa Fluor 647-conjugated anti-human IgA (JIR 109-605-011) for 30 min on ice, washed twice with cold PBS and measured on a Beckman Coulter CytoFLEX flow cytometer equipped with a 96-well plate reader.

A similar technique was used for testing the supernatants of spike-protein specific expanded B cells; in this case, 10 μl TE0/Tespike-cherry cells were incubated on ice with 15 μl of supernatant from each expanded B cell, in a 384-well plate (Thermo Scientific). After 30 min' incubation on ice, cells were transferred to a 96-well plate, washed twice in PBS, and labeled as described above.

## Surface plasmon resonance

SPR measurements were conducted on a Biacore T100 (T200 Sensitivity Enhanced, GE Healthcare Life Sciences). His-tagged spike protein or spike protein sub-domains (full-length extracellular domain, RBD, or S2; Sino Biological, catalog numbers 40589-V08H4, 40592-V08H, 40590-V08H1) were captured on a CM5 sensor chip (Cytiva) using the His Capture Kit, Type 2 (Cytiva) resulting in capture levels of 170 RU for Spike, 100 RU for S2, and 60 RU for RBD. All runs were conducted with HBS-P+ buffer (10 mM HEPES, 150 mM NaCl, 0.05% (v/v) Surfactant P20, pH 7.4; Cytiva). The different antibodies were injected in increasing concentrations (1.6, 8, 40, 200, 1,000 nM) for single cycle kinetics with a contact time of 120 s and dissociation time of 600 s at a flowrate of 30 μl/min. Dissociation constants (KD) were calculated with the BiacoreT200 Evaluation software 3.0 using the two-state-reaction model.

## VSV*ΔG-S$_{\Delta21}$ escape mutants

The selection of VSV*ΔG-S$_{\Delta21}$ mutants which escaped antibody-mediated inhibition was performed according to a recently described procedure (Walter et al, 2022). Briefly, approximately 100 focus-forming units of VSV*DG-SD21 were incubated with serially diluted IgM prior to infection of Vero E6 cells in 24-well plates. Two days later, cell culture supernatants from the wells containing the highest antibody concentration which did not completely inhibit virus plaque expansion was collected and subjected to a second round of selection on Vero E6 cells grown in 96-well microtiter

plates in the presence serially diluted IgM. Virus recovered after a third round of selection was used to infect Vero E6 cells grown in 6-well plates. Cells were lysed in GeneZol (GeneAid, New Taipei City, Taiwan) and total RNA isolated according to the manufacturer's instructions. Abundance of spike gene cDNA was quantified by an in-house quantitative PCR for SARS-CoV-2 (Leuzinger *et al*, 2021) and diluted to Ct = 20.

The diluted RNA was amplified and sequenced as described in Stange *et al* (2021) (Stange *et al*, 2021) with the following modifications: 35 cycles and v4.1 primers (https://github.com/artic-network/artic-ncov2019/tree/master/primer_schemes/nCoV-2019/V4.1) were used in the amplification.

Initial data analysis was done with the COVGAP pipeline (Stange *et al*, 2021).

### Statistics

Statistical treatments are explained in figure legends. We considered relevant sources of biological variation to include inter-individual differences between human donors, and genetic differences between individual virions. Possible sources of non-biological (i.e., technical) variability include experimental variables that vary from day to day, as well as from well to well, and so on. Therefore, rather than attempting to distinguish between "biological" and "technical" replicates, we have designed all experiments to cover both of these types of variability, and have explained the temporal and spatial structure in each figure legend. Experimental designs include single assays (e.g., virus neutralization by supernatants from single B cells); two or three temporal repeats (i.e., performed on different occasions) of single measurements (IgM and epitope ELISAS, flow cytometric measurements of antibody binding); and three temporal repeats of triplicate or quadruplicate technical replicates (neutralization assays). Statistics were calculated using GraphPad Prism. Plots were generated in R or in Prism, and compound figures were assembled using Inkscape.

## Data availability

Sequences of heavy and light chains of the antibodies shown in Appendix Table S3 are available in GenBank Accession numbers OM584288-OM584289, and OM687904-OM687941.

**Expanded View** for this article is available online.

### Acknowledgements
These studies were supported by the Swiss National Science Foundation (grant numbers 189043 and 189151). Technical support was provided by the flow cytometry, microscopy, and bioinformatics cores of the Department of Biomedicine of the University of Basel, and the Biophysics Facility of the Biozentrum of the University of Basel. We are grateful to Gennaro De Libero and John Lindner for technical and theoretical advice. Open access funding provided by Universitat Basel.

### Author contributions
**Ilaria Callegari:** Conceptualization; Data curation; Formal analysis; Investigation; Visualization; Methodology; Writing—original draft; Writing—review & editing. **Mika Schneider:** Investigation. **Giuliano Berloffa:** Formal analysis; Investigation; Methodology. **Tobias Mühlethaler:** Formal analysis; Investigation; Methodology. **Sebastian Holdermann:** Investigation; Methodology. **Edoardo Galli:** Software; Formal analysis; Visualization; Writing—review & editing. **Tim Roloff:** Data curation; Software; Investigation; Methodology; Writing—original draft; Writing—review & editing. **Renate Boss:** Investigation. **Laura Infanti:** Conceptualization; Data curation; Project administration. **Nina Khanna:** Resources; Data curation; Supervision. **Adrian Egli:** Conceptualization; Resources; Supervision; Project administration. **Andreas Buser:** Conceptualization; Resources; Supervision. **Gert Zimmer:** Conceptualization; Resources; Data curation; Formal analysis; Funding acquisition; Investigation; Methodology; Writing—original draft; Writing—review & editing. **Tobias Derfuss:** Conceptualization; Resources; Supervision; Funding acquisition; Methodology; Writing—original draft; Project administration; Writing—review & editing. **Nicholas S R Sanderson:** Conceptualization; Data curation; Software; Formal analysis; Validation; Investigation; Visualization; Methodology; Writing—original draft; Project administration; Writing—review & editing.

### Disclosure and competing interests statement
The authors declare that they have no conflict of interest.

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
