## [Review Process File · EMBO Reports]

Potent virus neutralization by human IgM against SARS-CoV-2 is impaired by class switch

Ilaria Callegari, Mika Schneider, Giuliano Berloff, Tobias Mühlethaler, Sebastian Holdermann, Edoardo Galli, Tim Roloff, Renate Boss, Laura Infanti, Nina Khanna, Adrian Egli, Andreas Buser, Gert Zimmer, Tobias Derfuss, and Nicholas Sanderson
DOI: [10.15252/embr.202153956](https://doi.org/10.15252/embr.202153956)

Corresponding author: Nicholas Sanderson (nicholas.sanderson@unibas.ch)

Review Timeline:

Submission Date:	7th Sep 21
Editorial Decision:	6th Oct 21
Revision Received:	5th Mar 22
Editorial Decision:	30th Mar 22
Revision Received:	13th Apr 22
Accepted:	27th Apr 22

Editor: Achim Breiling

Transaction Report:

Dear Dr. Sanderson,

Thank you for the submission of your research manuscript to EMBO reports. We have now received the reports from the three referees that were asked to evaluate your study, which can be found at the end of this email.

As you will see, the referees think that these findings are of interest. However, they have several comments, concerns and suggestions, indicating that a major revision of the manuscript is necessary to allow publication of the study in EMBO reports. As the reports are below, and all their points need to be addressed, I will not detail them here. During cross-commenting referee #1 agreed that a revision should be invited but stressed that all major/specific referee points need to be addressed experimentally.

Given the constructive referee comments, we would like to invite you to revise your manuscript with the understanding that all referee concerns must be addressed in the revised manuscript and in the detailed point-by-point response. Acceptance of your manuscript will depend on a positive outcome of a second round of review. It is EMBO reports policy to allow a single round of revision only and acceptance of the manuscript will therefore depend on the completeness of your responses included in the next, final version of the manuscript.

Revised manuscripts should be submitted within three months of a request for revision. Please contact me to discuss the revision should you need additional time.

1) a .docx formatted version of the final manuscript text (including legends for main figures, EV figures and tables), but without the figures included. Please make sure that changes are highlighted to be clearly visible. Figure legends should be compiled at the end of the manuscript text.

2) individual production quality figure files as .eps, .tif, .jpg (one file per figure), of main figures and EV figures. Please upload these as separate, individual files upon re-submission.

See also our guide for figure preparation:

4) a complete author checklist, which you can download from our author guidelines (<https://www.embopress.org/page/journal/14693178/authorguide>). Please insert page numbers in the checklist to indicate where the requested information can be found in the manuscript. The completed author checklist will also be part of the RPF.

5) that primary datasets produced in this study (e.g. RNA-seq, ChIP-seq, structural and array data) are deposited in an

appropriate public database. If no primary datasets have been deposited, please also state this a dedicated section (e.g. 'No primary datasets have been generated and deposited'), see below.

The accession numbers and database should be listed in a formal "Data Availability" section (DAS - placed after Materials & Methods) that follows the model below. This is now mandatory (like the COI statement). Please note that the Data Availability Section is restricted to new primary data that are part of this study.

Data availability

6) We strongly encourage the publication of original source data with the aim of making primary data more accessible and transparent to the reader. The source data will be published in a separate source data file online along with the accepted manuscript and will be linked to the relevant figure. If you would like to use this opportunity, please submit the source data (for example scans of entire gels or blots, data points of graphs in an excel sheet, additional images, etc.) of your key experiments together with the revised manuscript. If you want to provide source data, please include size markers for scans of entire gels, label the scans with figure and panel number, and send one PDF file per figure.

8) Regarding data quantification and statistics, can you please specify, where applicable, the number "n" for how many independent experiments (biological replicates) were performed, the bars and error bars (e.g. SEM, SD) and the test used to calculate p-values in the respective figure legends. Please provide statistical testing where applicable, and also add a paragraph detailing this to the methods section. See:
<http://www.embopress.org/page/journal/14693178/authorguide#statisticalanalysis>

9) Please also note our reference format:
<http://www.embopress.org/page/journal/14693178/authorguide#referencesformat>

10) Please add up to five keywords to the title page.

12) Please add a section to the manuscript text indicating the author contributions and a conflict-of-interest statement. Please order the manuscript sections like this:
Title page - Abstract - Introduction - Results - Discussion - Materials and Methods - DAS - Acknowledgements - Author contributions - Conflict of interest statement - References - Figure legends - Expanded View Figure legends

14) Finally, please add clear statements to the methods section indicating that all safety standards for SARS-CoV-2 work have been followed and the experiments have been approved by the respective institution.

I look forward to seeing a revised version of your manuscript when it is ready. Please let me know if you have questions or comments regarding the revision.

Yours sincerely,

Achim Breiling
Editor
EMBO Reports

Referee #1:

The manuscript by Callegari et al. investigated the IgM antibody response in convalescent individuals after SARS-CoV-2 infection or vaccinated donors. Therefore, the authors applied recombinant antibody technologies and isolated IgM sequences from B cells specific for the Spike of SARS-CoV-2 using their previously described MACAS technology (Zimmermann et al., 2019). These IgM were tested according to their neutralization potency and their ability to activate the complement system in comparison to IgGs and IgAs.

As key finding, the authors describe that neutralization of SARS-CoV-2 is dependent on isotype, meaning that IgM artificially converted to IgG lose their neutralization capacity. In general, this key finding is not new since it is known that the affinity matured IgM antibodies are of low affinity and that their pentameric structure increases avidity and thereby their efficiency e.g. to neutralize a virus. Of interest is the question of what affinity is sufficient for a single antibody and when B cells decide to switch classes. However, this important question is not investigated with scientific experiments in this manuscript.

In summary, the major finding of the manuscript is not significantly novel, the data is incomplete at several points and have severe technical weakness. Moreover, the conclusions which were drawn are not sufficiently supported by the experimental data. Therefore, in the current state, I cannot support publication of the manuscript in EMBO reports.

Major points:

- Figure 2 does not show the complete data set. The authors should at least include all experiments and data for all IgMs, IgGs and IgAs, which were used for the subsequently conducted experiments.
- In Figure 2C the authors claim that they have with 2E14 an epitope specific antibody, since the described SARS-CoV-2 spike variant was bound worse. Regarding the epitopes, the authors should perform experiments to investigate whether their different IgM antibodies have different or overlapping epitopes, e.g. by epitope binning. Also, further epitope analysis should be performed, e.g. by mutation studies or other technical approaches to confirm these statements. As presented, this assumption is not confirmed and, moreover, was analyzed only for one IgM.
- Antibodies used for these experiments were not purified but used as antibody containing supernatants for which concentrations were roughly determined by ELISA/ flow cytometry according to the Materials & Methods section. This is not standard practice in the antibody field since remaining cell supernatant proteins can have unknown influence (batch to batch variations) and analysis of protein integrity is only possible via Western Blot and not on isolated protein level.
- Throughout the manuscript the authors are referring to antibody affinity versus avidity and its influence on class switch e.g. "The results shown here suggest that a B cell ought not to switch from IgM to IgG until it has achieved a certain affinity through maturation, but do not explain how this decision is made". However, no affinity measurements were performed in the experimental setup, so such an assumption cannot be made. It would be very interesting to know and also necessary to show which KD for spike or RBD binding (determined for example by BLI or SPR) can be measured of the isolated Fabs from IgMs compared to Fabs from identified IgGs

Minor points:

- My general impression is, that in some places the text suffers from scientifically imprecise wording and unnecessary repetition of basic knowledge. An example is given in the first two sentences of the abstract (what means at some cost?). Introduction: "This pattern suggests that while IgG contributes most to enduring protection against re-infection, IgM and IgA must have properties that are particularly well-suited to immediate protection during an ongoing infection" Such basic facts are already fully acknowledged and not worth to be highlighted.
- In some places, the authors have made assumptions or statements that are not comprehensible. One example is given in the results section: "Firstly, using a cell-expressed protein as the assay antigen allows binding in a near-physiological conformation and orientation, and secondly the presence of all the endogenous cell membrane proteins enables assessment of polyreactive versus antigen-specific binding". In my opinion, this is a misleading assumption, since cells are not comparable to viruses.
- Figure legends (example):
Figure 1 "MACACS identifies spike-protein specific B cells of classes M, G, and A." What is shown here? Antibodies or antibody expressing B cells? To me the description is not sufficiently precise.
Figure 3 B: "Results like those shown on the left" All results should be shown (at least in the supplemental data).
- In many Graphs error bars (or replicates) are missing or like in Figure 2C it is not referred how many replicates were performed and how the data points are illustrated (mean {plus minus} s.d., etc.)
- The authors should include more recent literature especially comparing their work to other identified antibodies targeting SARS-CoV-2 isolated from patients/ vaccinated donors.
- The authors should show full sequences of the identified antibodies.

Referee #2:

Callegari et al investigated the properties of IgM antibodies after SARS-CoV-2 infection and COVID-19 vaccination. To this end,

the authors analyzed serum samples from SARS-CoV-2-experienced individuals, and they produced recombinant IgM antibodies as well as their IgG1 counterparts for binding and neutralization assays. They show that IgM antibodies contribute significantly to virus neutralization early in the immune response, which they attribute to their high avidity compared to IgG (and IgA). This is an interesting paper addressing an under-investigated topic concerning early antibody responses to virus infections, but several points require clarification.

Specific points:

Figure 1A: The authors find only low amounts of virus-specific IgM antibodies in their infected or vaccinated individuals, when testing serum samples in FACS using SARS-CoV-2 spike protein-expressing cells compared to non-transfected control cells. They explain their results by a higher background of IgM binding to control cells than in the experiments detecting IgG and IgA antibodies. In the left panel, however, it looks like that not only the unspecific binding was higher in their IgM assay, but also the specific binding seems to be lower. The authors should consider that their experimental approach might not be the most suitable for detecting IgM antibodies in serum, because virus-specific IgG and IgA antibodies have higher affinities/avidities than IgM antibodies. Therefore, these antibodies often outcompete IgM antibodies for antigens, making it difficult to reliably measure IgM levels in such assays. In addition, interference from rheumatoid factor IgM (that binds to IgG) could occur, leading to false-positive results.

It would be good to know the time point at which the samples of the infected/vaccinated individuals were obtained (only stated for 5 donors in Table 1) and whether they showed high IgM reactivity in classical diagnostic IgM assays.

Table 2: The term "some degree of neutralization" should be explained in a more scientific manner.

With the exception of Figure 2C, no error bars are shown throughout the paper (and they are not explained in the legend).

Figure 4A needs further explanations and it is not clear to me how the IgG data (blue symbols) shown in Figure 4A relate to the text. The authors describe that "at 1 µg/ml of each antibody IgG binding was eliminated", but I cannot see how the authors come to this conclusion with the data presented in Figure 4A.

The authors conclude from their data in Figure 4 that their results "comparing the binding of IgM and their derived IgG partners in competition are consistent with the hypothesis that the potency of IgM antibodies results from their avidity." It is also possible that the epitopes recognized by their recombinant IgM antibodies are less accessible to their IgG counterparts, because IgM have been described to be more flexible, thus being able to bind to epitopes that cannot be reached by the more rigid IgG antibodies. In addition, steric hindrance of IgG binding due to the larger IgM antibodies should also be considered in a competitive situation. Since no affinity/avidity measurements are presented in this paper, the authors should be more cautious in the interpretation of their results.

Referee #3:

In the present manuscript, Callegari et al. analyzed the effect of antibody isotype on SARS-CoV-2 neutralization. Using a system that detects antibody binding to TE cells expressing Spike (S) protein, they show that IgM from convalescent donors bind to TE cells regardless of S-protein expression, while IgG and IgA antibodies bind only to S-protein expressing cells. Attributing this to the polyreactive nature of IgM, they isolated S-binding B cells from the blood of convalescent donors based on their ability to extract S antigen from TE cells and to upregulate CD69 in this process. They single-cell sorted positive cells, activated them ex vivo and analyzed antigen binding of secreted antibodies. By this method, they found that CD69+ S-binding cells produced IgM antibodies specific for S-protein and not self-reactive. Remarkably, some of these monoclonal IgMs displayed the highest neutralization capacity and were able to activate complement, although there was no synergistic effect between complement and neutralization. Interestingly, when they artificially exchanged the Fc region of IgM neutralizing antibodies for an IgG, chimeric antibodies were outcompeted in their antigen binding capacity by native IgM antibodies and lost their neutralization capacity.

The present study provides compelling data on the importance of IgM antibodies in SARS-CoV-2 neutralization and offers some mechanistic clues. While I consider that these findings are valuable and should be published, substantial modifications are needed. In particular, authors should make a big effort to make the article more reader-friendly and highlight the novel aspects of their study, which in some sections are unclear.

Specific points:

1. Authors should specify the criteria used to select the 5 donors from Figure 1A for the subsequent antibody analysis in the rest of the study. Why are they all men? Which is their age?
2. In Figure 1A, They argue that "the lack of a stronger IgM signal on the spike-expressing cells is likely best explained by nonspecific IgM binding to the non-antigenic cells." This could be further attributed to the fact that the early IgM peak had already passed at the time point analyzed and detection is low (probably 1 month after infection? Not clear from the info provided). However, the IgM detected in Figure 1E is probably coming from the ex vivo differentiation of IgM+ S-specific memory

B cells circulating in blood. This point should be included in the text and discussed.

3. In figure 1D, authors should characterize better the CD69+ S-binding cells. Which is the composition of this population: Naive B, memory B, plasma cells?

4. Why do authors change the cut off from Figure 1A to 1E? Is the data presented in Figure 1E a pool of the 5 donors?

5.

- The authors show in Figure 2A the results for IgM and IgA but they don't show IgG, when they already mentioned it in the text. IgG data should be presented.

- They listed in table 2 the S-binding antibodies that showed "some degree of neutralization" (vague definition). They should include a comparison with those that are non-neutralizing and rigorous statistics.

- In Figure 2B authors should show the curves for the rest of the antibodies that they use later. Furthermore, they should quantify the neutralization capacity for IgA vs IgG vs IgM antibodies to assess whether differences are significant among isotypes.

6. According to Figure 2B, the clone 2J17 is more potent than 2E14. In fact 2E14 shows a similar neutralization pattern to the two IgA clones represented in the graph. Why do they use the 2E14 antibody for experiment in Figure 2C? Furthermore, analysis of IgM, IgG and IgA ability to neutralize SARS variants should be presented together with proper statistics to identify potential differences among isotypes. What are error bars? Mean plus DS? SEM? Median?

7. In Figure 2DTCID 50 is lower for 2E14 than 2J17, while apparently 2J17 is more neutralizing, is that difference significant?

8. It is textbook knowledge that IgM is the most effective isotype in activating complement, while IgG is less effective and IgA doesn't bind at all. Therefore, the novelty of panels 3A-C is not clear. Error bars are missing in Figure 3B.

9. Statistics are missing again in Figure 3D to conclude if the differences observed in the presence of complement are significant or not.

10. What do authors depict in Figure 4A (mean, median, error bars)? Why do authors use clone 3L11 that was not characterized in the previous figures? When authors change the IgM constant region for IgG, clones 3N8 and 2J17 lose binding capacity even without competition. Could this be explained by a change in conformation of the paratope?

11. In Figure 4C it is not clear whether neutralization was performed with the VSV or SARS technique. They depict the mean while they use the median for the rest of the graphs. At which concentrations do 2J17 and 2E14 lose neutralization capacity?

12. In Figure 4C, the authors attribute the neutralization capacity of S-specific antibodies to the IgM class. What happens if they change the Fc region of IgG or IgA neutralizing antibodies for the Fc of an IgM? Is the neutralization capacity increased?

13. It should be discussed if this phenomenon occurs only for SARS infection, or could happen in another viral infection like influenza for example.

14. A brief description of the samples should be included (in terms of age, gender, etc). At what time after infection the samples were obtained? How homogeneous is the distribution? How many cells were obtained for sorting? Which percentage of total cells represented the cells sorted?

15. In general, statistics are missing or poorly explained. Error bars are missing mostly in all the figures, and there is no consistency in the graphs among what is depicted (sometimes they depict the mean while others, the median).

Dear Editor, Dear Reviewers

Thank you for the invitation to submit a revised version of our manuscript

We are grateful for the three reviewers' very thorough consideration of the work, and the wide range of technical and theoretical insights they provided. We have completed all of the experiments suggested, and made all the proposed improvements to the presentation of the manuscript. A detailed description of the experiments conducted and the changes made follows each of the reviewers' points below.

Referee #1:

The manuscript by Callegari et al. investigated the IgM antibody response in convalescent individuals after SARS-CoV-2 infection or vaccinated donors. Therefore, the authors applied recombinant antibody technologies and isolated IgM sequences from B cells specific for the Spike of SARS-CoV-2 using their previously described MACAS technology (Zimmermann et al., 2019). These IgM were tested according to their neutralization potency and their ability to activate the complement system in comparison to IgGs and IgAs.

As key finding, the authors describe that neutralization of SARS-CoV-2 is dependent on isotype, meaning that IgM artificially converted to IgG lose their neutralization capacity. In general, this key finding is not new since it is known that the affinity matured IgM antibodies are of low affinity and that their pentameric structure increases avidity and thereby their efficiency e.g. to neutralize a virus. Of interest is the question of what affinity is sufficient for a single antibody and when B cells decide to switch classes. However, this important question is not investigated with scientific experiments in this manuscript.

In summary, the major finding of the manuscript is not significantly novel, the data is incomplete at several points and have severe technical weakness. Moreover, the conclusions which were drawn are not sufficiently supported by the experimental data. Therefore, in the current state, I cannot support publication of the manuscript in EMBO reports.

Major points:

- *Figure 2 does not show the complete data set. The authors should at least include all experiments and data for all IgMs, IgGs and IgAs, which were used for the subsequently conducted experiments.*

Neutralization data for all antibodies used in subsequent experiments are now shown in this figure (new Fig 4B, 4C).

- *In Figure 2C the authors claim that they have with 2E14 an epitope specific antibody, since the described SARS-CoV-2 spike variant was bound worse. Regarding the epitopes, the authors should perform experiments to investigate whether their different IgM antibodies have different or overlapping epitopes, e.g. by epitope binning. Also, further epitope analysis should be performed, e.g. by mutation studies or other technical approaches to confirm these statements. As presented, this assumption is not confirmed and, moreover, was analyzed only for one IgM.*

We have established by epitope binning that the neutralising IgM recognise independent epitopes (new Fig 7C). We have shown by studies of escape mutants that 2J17 interacts with the region around E486 in the RBD, and 2E14 with a well-described complex epitope in the NTD (new Fig 7D). As an additional confirmation of these results, we measured subdomain (S1, S2, NTD, RBD) binding for all antibodies studied by ELISA with recombinant subdomains (new Fig 7A). We extended the previous results with the del244-247/E484G variant to all six neutralising antibodies (new Fig 7B).

- *Antibodies used for these experiments were not purified but used as antibody containing supernatants for which concentrations were roughly determined by ELISA/ flow cytometry according to the Materials & Methods section. This is not standard practice in the antibody field since remaining cell supernatant proteins can have unknown influence (batch to batch variations) and analysis of protein integrity is only possible via Western Blot and not on isolated protein level.*

All antibodies have been purified by HPLC or protein G column purification (see Methods "Antibody purification")

- *Throughout the manuscript the authors are referring to antibody affinity versus avidity and its influence on class switch e.g. "The results shown here suggest that a B cell ought not to switch from IgM to IgG until it has achieved a certain affinity through maturation, but do not explain how this decision is made". However, no affinity measurements were performed in the experimental setup, so such an assumption cannot be made. It would be very interesting to know and also necessary to show which KD for spike or RBD binding (determined for example by BLI or SPR) can be measured of the isolated Fabs from IgMs compared to Fabs from identified IgGs*

We subjected all IgM and IgG antibodies to affinity measurement by Surface Plasmon Resonance, using spike or RBD or S2 as antigen (new Fig 6D). We were unable to find a method that reliably yielded Fab fragments from IgM, and so to make the comparison without the confound of valency, we made all the measurements with antibodies expressed as IgG1. We attempted to measure the Kd of whole IgM, but, as reported by Thouvenel et al. (PMID: 33661302) we were unable to do so. There were also 3 IgG for which we could not observe stable enough association and dissociation to calculate a reliable Kd in this paradigm.

Minor points:

- *My general impression is, that in some places the text suffers from scientifically imprecise wording and unnecessary repetition of basic knowledge. An example is given in the first two sentences of the abstract (what means at some cost?).*

Wording has been improved throughout the text to increase rigour and clarity.

Introduction: "This pattern suggests that while IgG contributes most to enduring protection against re-infection, IgM and IgA must have properties that are particularly well-suited to immediate protection during an ongoing infection" Such basic facts are already fully acknowledged and not worth to be highlighted.

We have removed this sentence and others that presented well-established immunological orthodoxy.

- *In some places, the authors have made assumptions or statements that are not comprehensible. One example is given in the results section: "Firstly, using a cell-expressed protein as the assay antigen allows binding in a near-physiological conformation and orientation, and secondly the presence of all the endogenous cell membrane proteins enables assessment of polyreactive versus antigen-specific binding". In my opinion, this is a misleading assumption, since cells are not comparable to viruses.*

We agree with the Reviewer's point that cells are not identical to viruses, and we have added measurements by ELISA to complement our cell-based approach (Fig 1D, Fig 7A and C, Fig EV2), and modified the explanation in the Results section (Page 3) accordingly.

- *Figure legends (example):*

Figure 1 "MACACS identifies spike-protein specific B cells of classes M, G, and A." What is shown here? Antibodies or antibody expressing B cells? To me the description is not sufficiently precise.

In the original submission, data concerning serum antibodies and the derivation of monoclonal antibodies from single B cells were presented on a single figure. We have separated the two kinds of results onto two separate figures (new Fig 1 and Fig 3) to make the distinction easier to follow. All the figure legends have been revised for clarity and completeness.

Figure 3 B: "Results like those shown on the left" All results should be shown (at least in the supplemental data). We have added binding figures for all of the antibodies described (new Fig EV1). The old Figures 3A and 3B in relation to complement were considered superfluous by Reviewer 3 and have been removed.

- *In many Graphs error bars (or replicates) are missing or like in Figure 2C it is not referred how many replicates were performed and how the data points are illustrated (mean {plus minus} s.d., etc.)*

Error bars have been added to all curves, and numbers of replicates and descriptions of data points have been included in all figure legends.

- *The authors should include more recent literature especially comparing their work to other identified antibodies targeting SARS-CoV-2 isolated from patients/ vaccinated donors.*

The Discussion has been extended to consider similarities and differences with anti-spike antibodies reported by other groups. Seventeen additional references have been included.

- *The authors should show full sequences of the identified antibodies.*

Full sequences of both chains of all antibodies have been uploaded to GenBank, and the accession numbers included in the manuscript. As of March 3rd, these sequences were not yet available publicly, and we have included GenBank flatfiles (Dataset EV1) with this submission for review purposes.

Referee #2:

Callegari et al investigated the properties of IgM antibodies after SARS-CoV-2 infection and COVID-19 vaccination. To this end, the authors analyzed serum samples from SARS-CoV-2-experienced individuals, and they produced recombinant IgM antibodies as well as their IgG1 counterparts for binding and neutralization assays. They show that IgM antibodies contribute significantly to virus neutralization early in the immune response, which they attribute to their high avidity compared to IgG (and IgA). This is an interesting paper addressing an under-investigated topic concerning early antibody responses to virus infections, but several points require clarification.

Specific points:

Figure 1A: The authors find only low amounts of virus-specific IgM antibodies in their infected or vaccinated individuals, when testing serum samples in FACS using SARS-CoV-2 spike protein-expressing cells compared to non-transfected control cells. They explain their results by a higher background of IgM binding to control cells than in the experiments detecting IgG and IgA antibodies. In the left panel, however, it looks like that not only the unspecific binding was higher in their IgM assay, but also the specific binding seems to be lower. The authors should consider that their experimental approach might not be the most suitable for detecting IgM antibodies in serum, because virus-specific IgG and IgA antibodies have higher affinities/avidities than IgM antibodies. Therefore, these antibodies often outcompete IgM antibodies for antigens, making it difficult to reliably measure IgM levels in such assays. In addition, interference from rheumatoid factor IgM (that binds to IgG) could occur, leading to false-positive results.

The idea that higher affinity, class switched antibodies in the serum might out-compete IgM in the context of our flow cytometric assay had not occurred to us. We looked at this possibility using a commercial anti-spike IgM ELISA based on the the method of first capturing human IgM from the serum onto the plate, and then detecting with a labeled recombinant spike RBD protein. As predicted by the mechanism described by the reviewer, this yields much clearer positives from the immunized population (new Fig 1C).

It would be good to know the time point at which the samples of the infected/vaccinated individuals were obtained (only stated for 5 donors in Table 1) and whether they showed high IgM reactivity in classical diagnostic IgM assays.

The time point information has been added to new Tables 1 and 2. High IgM reactivity in ELISA is indeed shown in Fig 1C.

Table 2: The term "some degree of neutralization" should be explained in a more scientific manner.

We apologise for this vague wording. To make this explanation clearer, we have added new Fig 3C, and clarified the corresponding explanation in the Results section (Page 4).

With the exception of Figure 2C, no error bars are shown throughout the paper (and they are not explained in the legend).

Error bars have been added to all curves, and their meaning specified in all figure legends.

Figure 4A needs further explanations and it is not clear to me how the IgG data (blue symbols) shown in Figure 4A relate to the text. The authors describe that "at 1µg/ml of each antibody IgG binding was eliminated", but I cannot see how the authors come to this conclusion with the data presented in Figure 4A.

The legend for this figure (new Fig 6A) and the corresponding explanation in the main text (Page 5) have been clarified.

The authors conclude from their data in Figure 4 that their results "comparing the binding of IgM and their derived IgG partners in competition are consistent with the hypothesis that the potency of IgM antibodies results from their avidity." It is also possible that the epitopes recognized by their recombinant IgM antibodies are less accessible to their IgG counterparts, because IgM have been described to be more flexible, thus being able to bind to epitopes that cannot be reached by the more rigid IgG antibodies. In addition, steric hindrance of IgG binding due to the larger IgM antibodies should also be considered in a competitive situation. Since no affinity/avidity measurements are presented in this paper, the authors should be more cautious in the interpretation of their results.

We have extended the consideration of other possible mechanisms to explain the class effect in the Discussion (Page 8). Affinity measurements are now presented in Fig 6D.

Referee #3:

In the present manuscript, Callegari et al. analyzed the effect of antibody isotype on SARS-CoV-2 neutralization. Using a system that detects antibody binding to TE cells expressing Spike (S) protein, they show that IgM from convalescent donors bind to TE cells regardless of S-protein expression, while IgG and IgA antibodies bind only to S-protein expressing cells. Attributing this to the polyreactive nature of IgM, they isolated S-binding B cells from the blood of convalescent donors based on their ability to extract S antigen from TE cells and to upregulate CD69 in this process. They single-cell sorted positive cells, activated them ex vivo and analyzed antigen binding of secreted antibodies. By this method, they found that CD69+ S-binding cells produced IgM antibodies specific for S-protein and not self-reactive. Remarkably, some of these monoclonal IgMs displayed the highest neutralization capacity and were able to activate complement, although there was no synergistic effect between complement and neutralization. Interestingly, when they artificially exchanged the Fc region of IgM neutralizing antibodies for an IgG, chimeric antibodies were outcompeted in their antigen binding capacity by native IgM antibodies and lost their neutralization capacity.

The present study provides compelling data on the importance of IgM antibodies in SARS-CoV-2 neutralization and offers some mechanistic clues. While I consider that these findings are valuable and should be published, substantial modifications are needed. In particular, authors should make a big effort to make the article more reader-friendly and highlight the novel aspects of their study, which in some sections are unclear.

We have removed unnecessary discussion of basic immunological concepts, and expanded and clarified the explanations of the work described and its novelty.

Specific points:

1. Authors should specify the criteria used to select the 5 donors from Figure 1A for the subsequent antibody analysis in the rest of the study. Why are they all men? Which is their age?

The donors were selected at random from among those with clear anti-spike antibodies in serum. This has been made clear in the Results section (Page 4). The donors were all men because the donations were collected in the context of study of the use of convalescent plasma for therapy of COVID-19. In Switzerland, and in many other countries, only men are eligible for plasma donation for this purpose, as part of a mitigation strategy against Transfusion Related Acute Lung Injury), on the grounds that parous women more frequently developed Anti-HLA and Anti-HNA Antibodies which are believed to be causative in immune TRALI.

This information about the origin of the samples has been added to the Methods section (Page 10, "Blood Samples") . The ages of the donors have been added to Table 2.

2. In Figure 1A, They argue that "the lack of a stronger IgM signal on the spike-expressing cells is likely best explained by nonspecific IgM binding to the non-antigenic cells." This could be further attributed to the fact that the early IgM peak had already passed at the time point analyzed and detection is low (probably 1 month after infection? Not clear from the info provided). However, the IgM detected in Figure 1E is probably coming from the ex vivo differentiation of IgM+ S-specific memory B cells circulating in blood. This point should be included in the text and discussed.

In response to a suggestion from Reviewer 2 we measured the anti-spike IgM by an IgM-specific ELISA, and the results (new Fig 1C) suggest that the exposed donors had significant levels of spike-specific IgM in serum that was underestimated by our flow cytometric assay. Information about the time between infection/vaccination and sample time point is now included in new Table 1, and indicated in the Results section (Page 3). The fact that the monoclonal antibodies are likely mostly derived from memory cells (see also next comment) has been highlighted in the Results section (last sentence of "SARS-CoV-2 spike protein-specific IgM B cells can be isolated by MACACS", Page 4).

3. In figure 1D, authors should characterize better the CD69+ S-binding cells. Which is the composition of this population: Naïve B, memory B, plasma cells?

We examined the phenotypes of CD69-high, spike-capturing B cells from 3 convalescent donors (new Fig 2), and the results make clear that these cells are preeminantly memory cells of IgM and IgG class (Results section Page 3-4).

4. Why do authors change the cut off from Figure 1A to 1E? Is the data presented in Figure 1E a pool of the 5 donors?

Originally the cutoff in the serum was set high to be conservative about which donors were considered "positive", while for the choice of which single B cell supernatants to analyse for spike-specific antibodies, we set the cutoff low to avoid missing any hits. We have reanalysed the data from Fig 1A using the absolute values of specific binding in an ANOVA, so that a cutoff is not necessary.

5.

- The authors show in Figure 2A the results for IgM and IgA but they don't show IgG, when they already mentioned it in the text. IgG data should be presented.

This has been added (new Fig 4A).

- They listed in table 2 the S-binding antibodies that showed "some degree of neutralization" (vague definition). They should include a comparison with those that are non-neutralizing and rigorous statistics.

We apologise for this confusing explanation. The description "some degree of neutralization" referred to results from the single cell expansion supernatants, and the Table described monoclonal antibodies. We have added new Fig 3C and clarified the corresponding explanation in the Results section " Monoclonal IgM mediate potent neutralization" (Page 4) to explain the selection of single cell expansion wells chosen for monoclonal antibody isolation. Information about derived monoclonals is presented in the new Table 3, and the graphical and statistical comparisons of neutralising and non-neutralising antibodies are now shown in Fig 4B and 4C.

- In Figure 2B authors should show the curves for the rest of the antibodies that they use later. Furthermore, they should quantify the neutralization capacity for IgA vs IgG vs IgM antibodies to assess whether differences are significant among isotypes.

All curves are now shown (new Fig 4B), and the statistical comparison forms new Fig 4C.

6. According to Figure 2B, the clone 2J17 is more potent than 2E14. In fact 2E14 shows a similar neutralization pattern to the two IgA clones represented in the graph. Why do they use the 2E14 antibody for experiment in Figure 2C? Furthermore, analysis of IgM, IgG and IgA ability to neutralize SARS variants should be presented together with proper statistics to identify potential differences among isotypes. What are error bars? Mean plus DS? SEM? Median?

This 2E14 was chosen for this experiment because of its apparently higher potency against the wild type SARS-CoV-2 (old Fig 2D, new Fig 4D). We measured the ability of 3 IgM, 2 IgA, and IgG to neutralise the del244-247/E484G variant. These were all the antibodies that showed strong neutralisation of the wild type spike. Results are shown in Fig 7B. The small number of antibodies involved does not offer the necessary statistical power to determine whether class has an effect on variant neutralisation.

7. In Figure 2D TCID₅₀ is lower for 2E14 than 2J17, while apparently 2J17 is more neutralizing, is that difference significant?

We compared the potency of 2J17 and 2E14 against wild type SARS-CoV-2 and against the SARS-CoV-2-pseudotyped VSV in three independent assays, and although in all three assays the 2E14 was more potent against wild virus, and less potent against the pseudotyped VSV, the difference was not statistically significant. This fact has been added to the legend of Fig 4.

8. It is textbook knowledge that IgM is the most effective isotype in activating complement, while IgG is less effective and IgA doesn't bind at all. Therefore, the novelty of panels 3A-C is not clear. Error bars are missing in Figure 3B.

We have removed Figures 3A and 3B, and compiled a new Fig 5, in which 5A corresponds to the old Fig 3C, and 5B is a complete version of the old 3D with appropriate error bars and statistics and a more explanatory legend.

9. Statistics are missing again in Figure 3D to conclude if the differences observed in the presence of complement are significant or not.

This has been corrected, please see previous comment.

10. What do authors depict in Figure 4A (mean, median, error bars)? Why do authors use clone 3L11 that was not characterized in the previous figures? When authors change the IgM constant region for IgG, clones 3N8 and 2J17 lose binding capacity even without competition. Could this be explained by a change in conformation of the paratope

The figure has been re-made with appropriate error bars, a full explanation of the meanings of points and axes, and inferential statistics. 3L11 has been removed to improve clarity. A brief consideration of the possible involvement of Fc influences on the antigen binding domain have been added to the Discussion (Page 8). We are cautious about making a direct comparison between binding of IgG and IgM in this paradigm, because the two classes of antibody are detected with different secondary antibodies using different fluoophores. When we made a similar comparison by ELISA (Fig EV2) we observed similar binding for the IgM-derived IgG as for their IgM parents.

11. In Figure 4C it is not clear whether neutralization was performed with the VSV or SARS technique. They depict the mean while they use the median for the rest of the graphs. At which concentrations do 2J17 and 2E14 lose neutralization capacity?

This figure has been fully corrected and forms the new Fig 6B and 6C. The new figure shows a wider range of antibody concentrations to make clear at what point the antibodies lose neutralisation capacity.

12. In Figure 4C, the authors attribute the neutralization capacity of S-specific antibodies to the IgM class. What happens if they change the Fc region of IgG or IgA neutralizing antibodies for the Fc of an IgM? Is the neutralization capacity increased?

We took two IgA and one IgG identified as neutralisers, switched them to IgM and examined their neutralising capacity (new Fig 6E). Results were heterogeneous. The weakly neutralizing IgG gained potency, while the originally most potent IgA lost potency.

13. It should be discussed if this phenomenon occurs only for SARS infection, or could happen in another viral infection like influenza for example.

This has been added to the Discussion (Page 8-9).

14. A brief description of the samples should be included (in terms of age, gender, etc). At what time after infection the samples were obtained? How homogeneous is the distribution? How many cells were obtained for sorting? Which percentage of total cells represented the cells sorted?

These data are now included in Tables 1 and 2.

15. In general, statistics are missing or poorly explained. Error bars are missing mostly in all the figures, and there is no consistency in the graphs among what is depicted (sometimes they depict the mean while others, the median).

Error bars have been added throughout, the graphs have been made consistent, and inferential statistics have been added as appropriate.

Dear Dr. Sanderson,

Thank you for the submission of your revised manuscript to our editorial offices. I have now received the reports from the three referees that were asked to re-evaluate your study, you will find below. As you will see, the referees now support publication of your work in EMBO reports. Nevertheless, all referees have remaining concerns, requests for clarification and suggestions to improve the manuscript I ask you to address in a final revised version of the manuscript. Please also provide a detailed p-b-p-response to these remaining points.

Moreover, I have these editorial requests I also ask you to address in a final revised manuscript:

- Please provide a title with not more than 100 characters (including spaces). See also the comment by referee #3.
- Please move the abstract to the second page of the manuscript (after the affiliations) and provide the abstract written in present tense.
- Please add up to 5 key words to the title page.
- We plan to publish your manuscript in the Report format. For a Scientific Report we require that results and discussion sections are combined in a single chapter called "Results & Discussion". Please do this for your manuscript. For more details please refer to our guide to authors: <http://www.embopress.org/page/journal/14693178/authorguide#researcharticleguide>
- Reports can have up to 5 main and 5 EV figures. Please present the data shown in the present 7 main figures and the two EV figures in 5 final main figures. I think it will be possible show all this data in 5 main figures. In case you require EV figures, please provide their legends in a section called Expanded View Figure Legends after the main Figure Legends section. If a figure has only 1 panel (as presently the EV figures), they don't need the "A" label. Please check (also the callouts).
- Please call out the figure panels sequentially (as they show up in the figure) or change their order in the figure. Presently panels 6E is called out before 6D.
- Please make sure that the number "n" for how many independent experiments were performed, their nature (biological versus technical replicates), the bars and error bars (e.g. SEM, SD) and the test used to calculate p-values is indicated in the respective figure legends (main and EV figures), and that statistical testing has been done where applicable. Please avoid phrases like 'independent experiment', but clearly state if these were biological or technical replicates. Please add complete statistical testing to all diagrams (main, EV and Appendix figures). Please also indicate (e.g. with n.s.) if testing was performed, but the differences are not significant.
- Please add a paragraph (termed Author contributions) detailing the contributions of each author to the manuscript text. Please place this before the acknowledgements.
- We updated our journal's competing interests policy in January 2022 and request authors to consider both actual and perceived competing interests. Please review the policy <https://www.embopress.org/competing-interests> and add a statement declaring your competing interests. Please name that section 'Disclosure and Competing Interests Statement' and add it after the acknowledgements section.
- The 'Data Availability' section (DAS) in the manuscript should contain links and accession numbers to datasets that have been deposited at external databases. If no large datasets have been submitted to a public database, the please just state there 'No large primary datasets have been generated and deposited'.
- I would suggest moving all the tables in an Appendix file. Please upload this as a single pdf file labeled Appendix. The Appendix should have page numbers and needs to include a table of content on the first page (with page numbers) and legends for all content. Please name the tables Appendix Table Sx and change their callouts in the manuscript text accordingly. Presently there are two tables named Table 5, and no Table 7 (although this is called out). Please check!
- Please select an answer in the field 'Laboratory protocol' of the author checklist.
- Walter et al. (EMBOR-2021-54199V3) mentioned as 'in press' in the Methods part has been published. Please update this and add a proper citation.
- Looking through the manuscript, I could not find anything regarding safety regulations, biosafety levels and institutional or governmental approval of the experiments. It seems you used live and mutant viruses. Please add a paragraph to the methods section (titled 'biosafety') providing details on that (clearly stating where and how biosafety-relevant experiments were performed and that these were approved).

- Finally, please find attached a word file of the manuscript text (provided by our publisher) with changes we ask you to include in your final manuscript text, and some queries, we ask you to address. Please provide your final manuscript file with track changes, in order that we can see any modifications done.

In addition, I would need from you:

- a short, two-sentence summary of the manuscript (not more than 35 words).
- two to four short bullet points highlighting the key findings of your study.
- a schematic summary figure (in jpeg or tiff format with the exact width of 550 pixels and a height of not more than 400 pixels) that can be used as a visual synopsis on our website.

Please use this link to submit your revision: Link Not Available

Yours sincerely,

Referee #1:

In their revision, Callegari et al addressed some of my requested points to my satisfaction. Nevertheless, more clarity is needed in some places.

Fig 1 D clearly showed that all exposed individuals have developed spike specific IgMs which were not detectable with methods used in Figure 1 A -C. Consequently the author should make a statement here and consider it for their further experiments.

Figure 2: CD69 is actually an activation marker for T cells not B cells. The authors should justify the use of CD69 here (e.g. by providing a reference for CD69 as a marker for activated B cells)

Figure 6 D: this is not an appropriate way to present results/data from SPR measurements. The authors should provide the corresponding sensograms at least in a EV figure.

Figure 7 B: The presentation of the virus neutralization as a table is incomprehensible from my point of view. The authors should present the corresponding neutralization curves and the IC50 values calculated from them.

In order to ensure comparability, I think that antibody concentrations should always be given as molar values and not as concentrations (ng or µg/ml).

Referee #2:

The authors provide answers to all of the reviewers' comments in their response letter as well as additional data, addressing my original concerns. However, the paper is still difficult to read and requires some clarifications.

Major points

- Page 5, section "IgM-mediated virus neutralization ...": The rationale for these experiments is unclear/vague. The authors used purified antibodies for their experiments, so why should "animal sera used at various points in the production and testing ... could provide complement precursors." I assume that the authors wanted to investigate whether neutralization of the different subclasses can be enhanced by complement (and they confirm textbook knowledge that IgM is most effective in this respect).
- The authors apparently used different types of pseudovirus neutralization assays (GFP, Luciferase) and a live-virus assay. Which type of assay was used should always be clearly stated in all legends and the "GFP/fluorescence-readouts" should also be described in Methods.

Minor points

- Line numbers would have been helpful
- Page 4, line 2: check wording (acquisition?)
- Page 4, end of first and second paragraph: some redundancy
- Page 5, section "Neutralization is dependent on class": It is unclear what the authors mean with "competitive paradigm"

(competition experiment? competitive setting?).

- Page 7, last para, second line: E486 should read E484, superepitope should read supersite.
- Page 7, last line: the E484K mutation is selected not induced.
- Legend Figure 4B: The x-axis shows the concentration of the antibodies and the y-axis a percentage (and not ratio).

Referee #3:

The authors took into account our comments, experimentally addressed our points and made substantial changes that improved the quality of the manuscript.

There are two points that I consider important:

- Fig.6A: The way that authors present the data leads the reader to automatically compare binding between IgM and IgG. Authors acknowledge in the point-by-point reply document, that we need to be careful when comparing this, as the antibodies used for their measurement are different. Then, these graphs should be depicted separately (one for each isotype "alone" vs "compet").
- The title is too strong. The authors claim that the neutralisation effect of IgM is eliminated when it changes to IgG based on the assay of only 3 antibodies (Fig 6B-C). This figure shows a reduction and not an elimination in neutralisation capacity. Furthermore, authors do not provide a mechanism for this phenomenon. I suggest to change the title to "neutralisation is reduced/impaired" instead of "is eliminated".

Referee #1:

In their revision, Callegari et al addressed some of my requested points to my satisfaction. Nevertheless, more clarity is needed in some places.

Fig 1 D clearly showed that all exposed individuals have developed spike specific IgMs which were not detectable with methods used in Figure 1 A -C. Consequently the author should make a statement here and consider it for their further experiments.

We have added the following statement regarding sensitivity and specificity of the flow cytometric and ELISA methods.

" The ELISA approach correctly identifies a higher fraction of exposed donors. The flow cytometric comparison between IgM binding to spike-expressing and non-expressing cells makes clear that some of the exposure-induced IgM binding activity is not virus-specific. Our data do not enable us to assess the quantitative tradeoffs between sensitivity and specificity, since this would require a more comprehensive cohort of control sera including donors exposed to other immunogenic pathogens. Infection with Epstein Barr Virus, for example, has been shown to induce spurious IgM RBD ELISA reactivity (Pickering *et al*, 2020)."

Figure 2: CD69 is actually an activation marker for T cells not B cells. The authors should justify the use of CD69 here (e.g. by providing a reference for CD69 as a marker for activated B cells)

The establishment of CD69 as an activation marker for the purpose of isolating antigen-specific B cells is described in detail in Zimmermann *et al.* (2019), cited in the manuscript. We have added a citation of a recent publication from an independent group (Malinova *et al.*, 2021, EMBO Rep. 2021 Sep 6; 22(9): e51328.)

Figure 6 D: this is not an appropriate way to present results/data from SPR measurements. The authors should provide the corresponding sensograms at least in a EV figure.

The sensograms have been added as Figure EV3.

Figure 7 B: The presentation of the virus neutralization as a table is incomprehensible from my point of view. The authors should present the corresponding neutralization curves and the IC50 values calculated from them.

Neutralization curves have been added as Figure EV4.

In order to ensure comparability, I think that antibody concentrations should always be given as molar values and not as concentrations (ng or µg/ml).

In general, we agree that molarities are the best way to express concentrations. In this particular case, after careful consideration, we elected to use mass per volume units, because in several places it is not clear whether the most meaningful molarity would be the molarity of the binding sites or the molarity of the multimeric molecules. The use of mass per volume units in antibody research, including the narrower context of SARS-CoV-2 neutralization, is well established (Robbiani *et al.*, Nature. 2020 584(7821): 437–442; Pinto *et al.* Science. 2021 373(6559):1109-1116; Suryadevara *et al.* Cell. 2021 184(9): 2316–2331).

Referee #2:

The authors provide answers to all of the reviewers' comments in their response letter as well as additional data, addressing my original concerns. However, the paper is still difficult to read and requires some clarifications.

Major points

•Page 5, section "IgM-mediated virus neutralization ...": *The rationale for these experiments is unclear/vague. The authors used purified antibodies for their experiments, so why should "animal sera used at various points in the production and testing ... could provide complement precursors." I assume that the authors wanted to investigate whether neutralization of the different subclasses can be enhanced by complement (and they confirm textbook knowledge that IgM is most effective in this respect).*

We apologize for the presence of this sentence in the new manuscript, which was a hangover from the previous version before antibody purification. It has now been removed.

•*The authors apparently used different types of pseudovirus neutralization assays (GFP, Luciferase) and a live-virus assay. Which type of assay was used should always be clearly stated in all legends and the "GFP/fluorescence-readouts" should also be described in Methods.*

We have clarified the section in Methods explaining the three types of neutralization assay used, by adding an introductory paragraph and using more consistent naming for the assays. We have also checked that each figure legend has the correct assay specification.

Minor points

•*Line numbers would have been helpful added*

•*Page 4, line 2: check wording (acquisition?) changed to "capture"*

•*Page 4, end of first and second paragraph: some redundancy*
We have shortened this section and removed some repetitive text.

•*Page 5, section "Neutralization is dependent on class": It is unclear what the authors mean with "competitive paradigm" (competition experiment? competitive setting?).*
changed to "competitive setting"

•*Page 7, last para, second line: E486 should read E484, superepitope should read supersite.*
corrected

•*Page 7, last line: the E484K mutation is selected not induced.*
corrected

•*Legend Figure 4B: The x-axis shows the concentration of the antibodies and the y-axis a percentage (and not ratio).*
corrected

Referee #3:

The authors took into account our comments, experimentally addressed our points and made substantial changes that improved the quality of the manuscript.

There are two points that I consider important:

-Fig.6A: The way that authors present the data leads the reader to automatically compare binding between IgM and IgG. Authors acknowledge in the point-by-point reply document, that we need to be careful when comparing this, as the antibodies used for their measurement are different. Then, these graphs should be depicted separately (one for each isotype "alone" vs "compet").

We have split the two classes into separate graphs.

-The title is too strong. The authors claim that the neutralisation effect of IgM is eliminated when it changes to IgG based on the assay of only 3 antibodies (Fig 6B-C). This figure shows a reduction and not an elimination in neutralisation capacity. Furthermore, authors do not provide a mechanism for this phenomenon. I suggest to change the title to "neutralisation is reduced/impaired" instead of "is eliminated".

"eliminated" replaced with "impaired".

Dr. Nicholas Sanderson
University of Basel
Switzerland

Dear Dr. Sanderson,

I am very pleased to accept your manuscript for publication in the next available issue of EMBO reports. Thank you for your contribution to our journal.

At the end of this email I include important information about how to proceed. Please ensure that you take the time to read the information and complete and return the necessary forms to allow us to publish your manuscript as quickly as possible.

As part of the EMBO publication's Transparent Editorial Process, EMBO reports publishes online a Review Process File to accompany accepted manuscripts. As you are aware, this File will be published in conjunction with your paper and will include the referee reports, your point-by-point response and all pertinent correspondence relating to the manuscript.

If you do NOT want this File to be published, please inform the editorial office within 2 days, if you have not done so already, otherwise the File will be published by default [contact: emboreports@embo.org]. If you do opt out, the Review Process File link will point to the following statement: "No Review Process File is available with this article, as the authors have chosen not to make the review process public in this case."

Thank you again for your contribution to EMBO reports and congratulations on a successful publication. Please consider us again in the future for your most exciting work.

Yours sincerely,

Achim Breiling
Editor
EMBO Reports

THINGS TO DO NOW:

Please note that you will be contacted by Wiley Author Services to complete licensing and payment information. The required 'Page Charges Authorization Form' is available here: https://www.embopress.org/pb-assets/embo-site/er_apc.pdf

You will receive proofs by e-mail approximately 2-3 weeks after all relevant files have been sent to our Production Office; you should return your corrections within 2 days of receiving the proofs.

Please inform us if there is likely to be any difficulty in reaching you at the above address at that time. Failure to meet our deadlines may result in a delay of publication, or publication without your corrections.

All further communications concerning your paper should quote reference number EMBOR-2021-53956V3 and be addressed to emboreports@wiley.com.

Should you be planning a Press Release on your article, please get in contact with emboreports@wiley.com as early as possible, in order to coordinate publication and release dates.